**Data Availability Statement:** Data cannot be shared publicly because of ethical concerns. However, data are available for researchers who meet the criteria for access to confidential data.

# Using visual attention estimation on videos for automated prediction of autism spectrum disorder and symptom severity in preschool children

Ryan Anthony J. de Belen [1]*, Valsamma Eapen[2], Tomasz Bednarz[3], Arcot Sowmya[1]

**1** School of Computer Science and Engineering, University of New South Wales, New South Wales, Australia, **2** School of Psychiatry, University of New South Wales, New South Wales, Australia, **3** School of Art & Design, University of New South Wales, New South Wales, Australia

* r.debelen@unsw.edu.au

## Abstract

Atypical visual attention in individuals with autism spectrum disorders (ASD) has been utilised as a unique diagnosis criterion in previous research. This paper presents a novel approach to the automatic and quantitative screening of ASD as well as symptom severity prediction in preschool children. We develop a novel computational pipeline that extracts learned features from a dynamic visual stimulus to classify ASD children and predict the level of ASD-related symptoms. Experimental results demonstrate promising performance that is superior to using handcrafted features and machine learning algorithms, in terms of evaluation metrics used in diagnostic tests. Using a leave-one-out cross-validation approach, we obtained an accuracy of 94.59%, a sensitivity of 100%, a specificity of 76.47% and an area under the receiver operating characteristic curve (AUC) of 96% for ASD classification. In addition, we obtained an accuracy of 94.74%, a sensitivity of 87.50%, a specificity of 100% and an AUC of 99% for ASD symptom severity prediction.

## Introduction

Autism spectrum disorders (ASD) are currently being diagnosed through visual observation and analysis of children's natural behaviours. While a gold standard observational tool is available, early screening of ASD in children still remains a complex problem. It is often expensive and time-consuming [1] to conduct interpretative coding of child observations, parent interviews and manual testing [2]. In addition, differences in professional training, resources and cultural context may affect the reliability and validity of the results obtained from a clinician's observations [3]. Furthermore, the behaviours of children in their natural environments (e.g., home) cannot be typically captured by clinical observation ratings. To reduce waiting periods for access to interventions, it is important to develop new methods of ASD diagnosis without compromising accuracy and clinical relevance. This is critical because early diagnosis and

The eye-tracking data is available in a de-identified format upon request as imposed by the UNSW Human Research Ethics Committee. The clinical data cannot be published since the UNSW Human Research Ethics Committee and the families have not given us permission for the use of data for secondary analysis. Data access requests can be sent to the Ingham Institute (bestart@inghaminsititute.org.au).

**Funding:** VE received funding from the Cooperative Research Centre for Living with Autism (Autism CRC) (established and supported under the Australian Government's Cooperative Research Centre Program) for the staff and non-staff in kind that supported the collection of the clinical data used in this study. The funders had no role in study design, data collection and analysis, decision to publish, or preparation of the manuscript.

**Competing interests:** The authors have declared that no competing interests exist.

intervention can provide long-term improvements for the child and even have a greater effect on clinical outcomes [4].

Recent advances in technology have allowed for the quantification of different biological and behavioural markers that are useful in ASD research (see [5,6] for reviews). Eye-tracking technology has shown promise in providing a non-invasive and objective tool for ASD research [7,8]. Several eye-tracking studies have identified unique visual attention patterns in ASD individuals. Gaze abnormalities in toddlers (<3-year-olds) include reduced attention to eye and head regions, reduced preference for biological motion, difficulties in response to joint attention behaviours [9] and scene monitoring challenges during explicit dyadic cues [10]. Pierce, et al. [11], Pierce, et al. [12] and Moore, et al. [13] developed a geometric preference ("GeoPref") test that contains both geometric and social videos. It was found that a subset of ASD participants exhibited a visual preference for geometric motion. This finding has already been leveraged by a growing number of studies that aim to leverage atypical visual attention to identify individuals with ASD [14,15] and predict symptom severity [16].

Computational models that predict visual attention (i.e., saliency) have seen tremendous progress, starting from handcrafted features dating back to 1998 [17] to a resurgence of deep neural networks (DNNs) [18,19]. This breakthrough has generated great interest in utilising saliency prediction as a diagnostic paradigm for ASD. For example, there is a growing collection of eye movements of ASD children recorded during image-[20–22] and video-[22] viewing tasks. Although the use of saliency detection models on image datasets has resulted in remarkable diagnostic performance, there is still a lack of diagnostic paradigms that utilise dynamic saliency detection. In fact, the most common approach of studies that utilise dynamic stimuli is to convert the eye-tracking data into an image and perform image classification to identify individuals with ASD. In this work, we present a novel pipeline that leverages the dynamic visual attention of humans for ASD diagnosis, as well as symptom severity prediction.

This paper makes three major contributions to the field. First, we implement a data-driven approach to learn the dynamic visual attention of humans on videos and extract spatiotemporal features for downstream tasks (e.g., ASD classification and symptom severity prediction). Second, we develop a novel computational pipeline to diagnose ASD based on the learned features from dynamic visual stimuli. Finally, we use a similar method to predict the level of ASD-related symptoms from eye-tracking data of children obtained during a free-viewing task. In the next section, we discuss published works that are related to ours. Despite the growing literature, it is evident that the comparison of results is challenging due to the lack of publicly available datasets and open-source code repositories. This is even further complicated by the differences in the participants, age group and stimuli used in the experiments, making fair and straightforward performance comparisons more difficult. Nevertheless, we compare our work with a simple thresholding technique [11–13] and a machine learning (ML) classification approach using handcrafted features [23,24].

## Related works

Over the last decade, different behavioural and biological markers have already been quantified, to some extent, using computer vision methods (a comprehensive review [5] is available). Various data modalities, such as magnetic resonance imaging (MRI)/functional MRI [25–30], eye-gaze data [14,31–36], stereotyped behaviours [37–42] and multimodal data [43] have been utilised in autism diagnosis. We first provide a review of publicly available datasets that utilise the eye-tracking paradigm. Afterwards, related works that utilise eye-tracking data for the following purposes are reviewed: (i) saliency prediction in ASD, (ii) ASD diagnosis using static

stimuli, (iii) ASD diagnosis using dynamic stimuli and (iv) ASD risk and symptom severity prediction. Each purpose has a corresponding table that includes the following information about the published research: mean age of the participants, gender distribution, stimuli and input used, methodology and conclusion. While not as exhaustive and rigorous in inclusion criteria as a systematic review, we hope that our discussion below will help the readers navigate the research landscape and better situate our work in the literature. Readers are also encouraged to read systematic reviews [8,44] for additional reference.

## Publicly available datasets

There is a growing number of publicly available datasets that capture the eye-tracking data of ASD participants. In Table 1, we provide a summary of these datasets by providing descriptions of their target application area, the mean age of the participants, sample size, stimuli used and data format provided by the authors. There are two datasets for saliency estimation [20,21] and two datasets for ASD classification [22,45].

## Saliency prediction in ASD

Accurately predicting the visual attention (i.e., saliency maps) of ASD individuals can boost prediction performance because classification models can better leverage the distinction between the visual attention of ASD and typically developing (TD) individuals. Table 2 shows the published research that aims to model the visual attention of ASD participants by developing different saliency models.

Duan, et al. [46] compared the performance of five state-of-the-art (SOTA) saliency prediction networks based on a deep neural network (DNN) architecture with pre-trained and fine-tuned weights on their dataset. Experimental results revealed that transfer learning provides a useful approach to modelling visual attention on images for individuals with ASD. Duan, et al. [47] combined high-level features (e.g., face size, facial features, face pose and facial expressions) and feature maps extracted from the SOTA saliency models to quantify visual attention on human faces in ASD. Their proposed approach reported higher performance when compared to other saliency models.

The remaining works used the Saliency4ASD dataset [20,21] for saliency estimation. For example, Fang, et al. [48] used U-net trained on a novel loss function for semantic feature

**Table 1. List of publicly available datasets and their corresponding application area, mean age, sample size, stimuli and data format provided by the authors.**

| Authors | Application area | Mean age (SD) in years | Sample size | Stimuli | Data format |
|---|---|---|---|---|---|
| Duan, et al. [20] and Gutiérrez, et al. [21] (Saliency4ASD dataset) | Saliency estimation ASD classification | All participants: 8.00 (NR) | ASD: 14 TD: 14 | 300 images that depict diverse naturalistic scenes and may contain humans, animals, buildings or objects. | Image with the associated eye-tracking data of the participants |
| Le Meur, et al. [22] (MIE Fo and MIE No) | Saliency estimation | MIE Fo: ASD: 16.00 (2.00) MIE No: 29.00 (7.00) | MIE Fo: ASD: 17 MIE No: ASD: 12 | 25 images with low semantic meaning and a low emotional arousal | Image with the associated eye-tracking data of the participants |
| Carette, et al. [45] | ASD classification | All participants: 7.88 (NR) | ASD: 29 TD: 30 | Combination of static and dynamic stimuli that depict naturalistic scenes, initiate joint attention and static face or objects | Scanpath image that visualises the eye-tracking data of the participants. The visualised scanpath images are then converted to grayscale and rescaled for further processing. |

ASD: Autism Spectrum Disorder, NR: Not reported, SD: Standard deviation, TD: Typically Developing

**Table 2. Saliency prediction in ASD.**

| Authors | Mean age (SD) in years | Sample size | Stimuli | Input used | Method | Conclusions |
|---|---|---|---|---|---|---|
| Duan, et al. [46] | 7.8 (NR) | 13 | 500 images | Image | They compared the performance of five different SOTA saliency models. | Transfer learning provides a useful approach to model the visual attention on images in individuals with ASD. |
| Duan, et al. [47] | ASD: 7.80 (2.10) TD: 8.00 (2.00) | ASD: 13 TD: 15 | VAFA dataset: 300 images from open-source dataset [53] that depict various emotions and then classified into six expressions: (generally positive, very positive, neutral, generally negative, very negative and complex expressions) | Image | They computed fixation distributions on different pre-defined AOIs. Afterwards, statistical analyses were performed to identify differences in visual attention of ASD and TD participants while looking at effects of face pose and facial expressions. Afterwards, they compared six different SOTA deep learning-based saliency models on the VAFA dataset. | CASNet achieved the best performance in terms of the prediction of atypical visual attention of ASD individuals. |
| Fang, et al. [48] | Saliency4ASD | Saliency4ASD | Saliency4ASD | Image | They developed a saliency model based on the U-Net architecture. They also designed a new loss function called Positive and Negative Equilibrium Mean Square-Error that is used to determine model convergence. | Their model achieved higher performance on some metrics when compared to general saliency models. |
| Wei, et al. [49] | Saliency4ASD | Saliency4ASD | Saliency4ASD | Image | They first extracted multi-level features and combined these features using a fusion layer to output a saliency map. Deep supervision on the predicted saliency map was implemented to train the deeper layers of the network. They also utilised a single-side clipping approach to highlight regions that are mostly viewed by the participants. | Their model achieved the best performance on different metrics when compared to general saliency models. |
| Nebout, et al. [50] | Saliency4ASD | Saliency4ASD | Saliency4ASD | Image | They developed a two-stream network that extracts fine-scale and contextual information from the input image and the downscaled input image, respectively. Afterwards, a series of convolutional operations and concatenation is implemented to generate the saliency map. | Their model achieved the best performance on most metrics when compared to general saliency models. |
| Fang, et al. [51] | Saliency4ASD | Saliency4ASD | Saliency4ASD | Image | They modelled the dynamic nature of human visual attention using a two-stream model that consists of a CNNs and a series of convolutional LSTM layers. | Their model achieved the best performance on most metrics when compared to general saliency models and ASD-specific saliency models [48–50]. |
| Wei, et al. [52] | Saliency4ASD | Saliency4ASD | Saliency4ASD | Image | They first extracted multi-level features from the input image. Afterwards, they passed it to a scale-adaptive coarse-and-fine inception module for a richer representation. These features are then combined using a feature fusion module and passed to a refinement and integration module. To better learn the atypical visual attention of ASD individuals, they developed a discriminative region enhancement loss. | Their approach achieved the best performance on different metrics when compared to general saliency models and ASD-specific saliency models [48–50]. Their experiments showed that their novel loss function improved the performance of other models in predicting atypical visual attention of ASD participants. |

*(Continued)*

**Table 2.** (Continued)

| Authors | Mean age (SD) in years | Sample size | Stimuli | Input used | Method | Conclusions |
|---------|------------------------|-------------|---------|------------|--------|-------------|
| Le Meur, et al. [22] | Saliency4ASD MIE Fo and MIE No | Saliency4ASD MIE Fo and MIE No | Saliency4ASD MIE Fo and MIE No | Image | They compared six different saliency prediction models and analyse their saliency prediction performance in Saliency4ASD, MIE Fo and MIE No datasets. | Their results showed that current saliency models do not generalise well on ASD-specific dataset, hoping to raise awareness that researchers need different approaches to model the atypical visual attention of ASD people. |

AOI: Area Of Interest, ASD: Autism Spectrum Disorder, LSTM: Long Short-Term Memory, NR: Not reported, SD: Standard deviation, SOTA: State-of-the-art, TD: Typically Developing

learning, resulting in improved performance on some metrics. Wei, et al. [49] proposed a novel saliency prediction model for children with ASD. The fusion of multi-level features, deep supervision on attention maps and the single-side clipping operated on ground truths provided a boost in saliency prediction. Nebout, et al. [50] proposed a Convolutional Neural Network (CNN) with a coarse-to-fine architecture and trained using a novel loss function, achieving the best performance on most metrics when compared to general saliency models. Fang, et al. [51] proposed a model consisting of a spatial feature module and a pseudo-sequential feature module to generate an ASD-specific saliency map. Their model achieved the best performance on most metrics when compared to general saliency models and ASD-specific saliency models [48–50]. Finally, Wei, et al. [52] proposed a DNN architecture that enhances multi-level side-out feature maps using a scale-adaptive coarse-and-fine inception module. In addition, they designed a novel loss function to fit the atypical pattern of visual attention, resulting in SOTA performance.

This growing evidence suggests that researchers are starting to develop computational models that mimic the atypical visual attention on images of ASD individuals. However, there is still a huge gap in prediction performance as saliency prediction models trained on TD individuals do not generalise well on ASD individuals, as highlighted by Le Meur, et al. [22]. They revealed that current models trained on a TD dataset and fine-tuned on an ASD dataset perform well only on a small part of the ASD spectrum. To this end, they proposed two new eye-tracking datasets that cover a large part of the ASD spectrum.

## Eye-tracking on static stimuli for ASD diagnosis

As discussed in the previous section, it has been found that ASD participants exhibit atypical visual attention. As shown in Table 3, researchers explored the possibility of using the eye-tracking paradigm during image-viewing tasks to identify individuals with ASD. The earliest works explored different handcrafted features and ML models for ASD diagnosis. For example, Wang, et al. [54] used features extracted from images followed by a Support Vector Machine (SVM), while Yaneva, et al. [55] explored logistic-regression classification algorithms for detecting high-functioning ASD in adults. Liu, et al. [34] proposed a ML framework based on the frequency distribution of eye movements recorded during a face recognition task to identify individuals with ASD. The recent advances in deep learning (DL) also helped researchers better extract discriminative features from images. For example, Jiang and Zhao [33] used a DL approach followed by an SVM to distinguish individuals with ASD.

The succeeding works used the Saliency4ASD dataset [20,21]. Startsev and Dorr [56] and Arru, et al. [57] extracted features from the eye-tracking data and the input image and trained

**Table 3. Eye tracking on static stimuli for ASD diagnosis.**

| Authors | Mean age (SD) in years | Sample size | Stimuli | Input used | Method | Conclusions |
|---|---|---|---|---|---|---|
| Wang, et al. [54] | ASD: 30.80 (11.1)<br>TD: 32.30 (10.40) | ASD: 20<br>TD: 13 | 700 images from the OSIE dataset | Pixel-, object-, and sematic-level features extracted from the image. In addition, the image centre and background, as well as the ground-truth fixation maps were used. | Using the extracted features, they implemented an SVM to generate feature weights that were then combined to predict human fixation maps. They also conducted statistical analysis to investigate the atypical visual attention of ASD participants. | Their approach reported high performance in predicting the visual attention of both ASD and TD group. Their results showed that ASD group had increased biased towards the image centre, background and pixel-level, but reduced biased towards objects and semantic content of the image. |
| Yaneva, et al. [55] | Study 1:<br>ASD: 37.00 (9.14)<br>TD: 33.60 (8.60)<br>Study 2:<br>ASD: 41.00 (14.00)<br>TD: 32.20 (9.90) | Study 1:<br>ASD: 15<br>TD: 15<br>Study 2:<br>ASD: 19<br>TD: 19 | Study 1:<br>6 webpages with increasing visual complexity (e.g., low, medium, high) and 2 webpages in each category.<br>Study 2:<br>8 randomly selected webpages from a list of top 100 websites, ensuring that there are 4 low visual complexity and 4 high visual complexity content. | Different computed eye-tracking variables (e.g., number of fixations, time to first look at an AOI) and non eye-tracking data-related variables (e.g., gender, visual complexity) | They computed eye-tracking related variables on different pre-defined AOIs. Afterwards, they trained several logistic regression classifiers using different combinations of the feature set for ASD classification. | Their results suggest that atypical visual attention of ASD individuals can be used as a biomarker for classification. They found differences in the information processing of ASD participants, regardless of specific information-location instructions across different time conditions.<br>They also found that stimuli content and granularity have an impact on classification accuracy, while the stimuli complexity and gender do not exhibit the same effect. |
| Liu, et al. [34] | ASD: 7.90 (1.45)<br>TD-Age Matched: 7.86 (1.38)<br>TD-IQ Matched: 5.74 (1.01) | ASD: 29<br>TD-Age Matched: 29<br>TD-IQ Matched: 29 | 12 photos of adult Chinese female faces and 12 Caucasian female faces. 6 were used for memorisation task and 18 were used for a recognition task of the 6 memorised faces. | Frequency distribution of the visual attention of participants were computed. | They first quantised the fixation distribution of all participants using the k-means algorithm to generate cluster centroids. Afterwards, given a sequence of fixation locations, they assigned the cluster centroid closest to a participant's fixation location and counted the frequency of cluster assignments. This process was repeated on all the images and an SVM classifier was used for classification. | Their results showed a promising performance in classifying ASD participants based on visual attention on human faces. |

*(Continued)*

**Table 3.** (*Continued*)

| Authors | Mean age (SD) in years | Sample size | Stimuli | Input used | Method | Conclusions |
|---------|------------------------|-------------|---------|------------|--------|-------------|
| Jiang and Zhao [33] | Same as Wang, et al. [54] | Same as Wang, et al. [54] | Same as Wang, et al. [54] | Images (and corresponding rescaled images) with the associated eye-tracking data of the participant | First, image selection using Fisher score ranking was implemented to reduce the number of input images from 700 to 100. Afterwards, each image and it corresponding rescaled image were passed to a two branch VGG-16 network. The extracted features were then concatenated and used to predict the difference of fixation maps. Afterwards, a latent representation in the model was used for classification using SVM. | There was no direct comparison with other models since their model was one of the first to use eye-tracking for ASD classification. Nevertheless, the authors compared their approach with similar work that used different group of subjects and input data and received the highest performance across different metrics. |
| Startsev and Dorr [56] | Saliency4ASD | Saliency4ASD | Saliency4ASD | Images with the associated eye-tracking data of the participant, including fixation durations. | First, they computed features extracted from the eye-tracking data and the input image. Afterwards, they trained a random forest for classification. | Their analysis revealed that images that contain multiple faces provide significant differences in visual attention between ASD and TD individuals. |
| Wu, et al. [58] | Saliency4ASD | Saliency4ASD | Saliency4ASD | Images with the associated eye-tracking data of the participant, including fixation durations. | They developed two networks: Synthetic saccade approach: a synthetic data generated by a scanpath model is aligned with the real eye-tracking data. Distance measures were then computed on these two data. Afterwards, different eye-tracking statistics were concatenated and used as features for MLP classification. Image-based approach: the real eye-tracking data were converted into an image. Afterwards, features were extracted from the input stimulus and the converted image and used as features for classification. | Their experiments showed that both approaches resulted in similar classification performance in terms of accuracy and AUC. |
| Arru, et al. [57] | Saliency4ASD | Saliency4ASD | Saliency4ASD | Images with the associated eye-tracking data of the participant, including fixation durations. | First, they extracted features extracted from the image, eye-tracking data and bias towards the image centre. Afterwards, they trained a random forest that uses a bagging algorithm for classification. | Their results suggested that scene analysis, such as determining the objects attended by participants, could provide better results. |

(*Continued*)

**Table 3.** (Continued)

| Authors | Mean age (SD) in years | Sample size | Stimuli | Input used | Method | Conclusions |
|---|---|---|---|---|---|---|
| Tao and Shyu [59] | Saliency4ASD | Saliency4ASD | Saliency4ASD | Images with the associated eye-tracking data of the participant, including fixation durations. | First, they used a saliency model to generate a saliency map for a given image. Afterwards, square patches centred around the participant's fixations were extracted from the predicted saliency map. These patches were then passed to a CNN for feature extraction. The gaze duration associated with a patch location is concatenated with the extracted patch features and sequentially passed to an LSTM network followed by an FCL for classification. | Their results achieved an accuracy of 74.22% on the validation set and 57.90% on the test set. |
| Chen and Zhao [43] | Photo-taking task: NR Image-viewing task: NR Saliency4ASD | Photo-taking task: ASD: 22 TD: 23 Image-viewing task: ASD: 20 TD: 19 Saliency4ASD | Photo-taking task: First-person photo taken by the participant Image-viewing task: 700 images from the OSIE dataset Saliency4ASD | Photo-taking task: First-person photo taken by the participant Image-viewing task: Images with the associated eye-tracking data of the participant. Saliency4ASD | Photo-taking task: Given a sequence of photos taken by the participant, features are extracted using a CNN network and passed into a global average pooling layer. The sequence of image features is passed into an LSTM network and an FCL for classification. Image-viewing task: Given an image, features are extracted using a CNN network. Afterwards, using the associated eye-tracking data, features are extracted around the fixation location. The sequence of extracted features is then passed into an LSTM network and a FCL for classification. The authors also used multi-modal distillation to train both models. | Their results had the highest accuracy performance when compared to other models [33,34]. |

(*Continued*)

**Table 3.** (Continued)

| Authors | Mean age (SD) in years | Sample size | Stimuli | Input used | Method | Conclusions |
|---|---|---|---|---|---|---|
| Fang, et al. [60] | Saliency4ASD GazeFollow4ASD: ASD: 9.60 (NR) TD: 8.90 (NR) | Saliency4ASD GazeFollow4ASD: ASD: 8 TD: 10 | Saliency4ASD GazeFollow4ASD: Images that contain people looking at other people/objects in the scene | Saliency4ASD GazeFollow4ASD: Images with the gaze-following prior map indicating the eye locations of the people in the image and their gaze locations | First, they used a dilated CNN to extract coarse feature maps from the input image. Afterwards, these feature maps are passed to a convolutional LSTM network to generate enhanced features. A fusion layer is used to add the gaze-following prior map and a series of CNN layers is used to generate a difference of fixation maps. A latent representation in the model is passed to two FCLs for classification. | Their results had the highest accuracy performance when compared to a model [33] submitted to Saliency4ASD. |
| Rahman, et al. [61] | Saliency4ASD | Saliency4ASD | Saliency4ASD | Images with the associated eye-tracking data of the participant. | First, they used seven different saliency prediction models on a given image and computed evaluation metrics between the predicted saliency and the recorded eye tracking data of the participant. This process is repeated for all the viewed images. The evaluation results for each saliency prediction model were concatenated. This feature representation was passed to an SVM and XGBoost for comparison of classification performance. | Their model reported a higher performance compared to a previous SOTA model [43] for ASD classification. |
| Xu, et al. [62] | Saliency4ASD | Saliency4ASD | Saliency4ASD | Images with the associated eye-tracking data of the participant. | Using structural similarity, they selected a subset of images that resulted into significant differences in visual attention of ASD and TD participants. Afterwards, they developed a bio-inspired metric that classifies ASD using the eye-tracking data. | Their results suggest that screening the photos to be viewed by participants and eventually used for classification is necessary to increase the model accuracy. |

(*Continued*)

**Table 3.** (Continued)

| Authors | Mean age (SD) in years | Sample size | Stimuli | Input used | Method | Conclusions |
|---|---|---|---|---|---|---|
| Wei, et al. [63] | Saliency4ASD | Saliency4ASD | Saliency4ASD | Images with the associated eye-tracking data of the participant. | First, an image encoder was used to extract visual features. Afterwards, the associated eye-tracking data of the participant was used as an input to three branches: (1) embedding layer to extract features (2) field of view maps generator layer that is composed of a spatial attention mechanism and LSTM network to extract spatiotemporal features (3) dynamic filters generator layer that uses CNNs. A final two FCLs were used for classification. | Their results had the highest accuracy and similar specificity and AUC scores when compared to models [56–59] submitted to Saliency4ASD. |
| Liaqat, et al. [64] | Saliency4ASD | Saliency4ASD | Saliency4ASD | Images with the associated eye-tracking data of the participant | They developed two networks: Branched MLP approach: it consists of a three-branch network that processes three different kinds of features: (1) a synthetic saccade is generated using a scanpath model, (2) a real scanpath and (3) statistical features. These features are passed to a series of MLPs for classification. Image-based approach: it consists of a two-branch network that extracts features from the input image and the eye tracking data and uses a final classification layer. | The image-based approach resulted in slightly better results than the branched MLP approach. |
| Mazumdar, et al. [65] | Saliency4ASD | Saliency4ASD | Saliency4ASD | Images with the associated eye-tracking data of the participant. | They computed features extracted from the image, eye-tracking data and centre bias of participants. Afterwards, they trained 23 different classifiers, such as decision trees, naïve bayes classifier, SVM, nearest neighbour classifier, and ensemble-based classifiers. | Their results were among the top 4 performing models across different metrics when compared to models [56,59,64] submitted to Saliency4ASD. |

AOI: Area of Interest, ASD: Autism Spectrum Disorder, AUC: Area Under the Curve, CNN: Convolutional Neural Network, FCL: Fully-Connected Layer, IQ: Intelligence Quotient, LSTM: Long Short-Term Memory, MLP: Multi-Layer Perceptron, NR: Not reported, SD: Standard deviation, SOTA: State-Of-The-Art, SVM: Support Vector Machine, TD: Typically Developing

a random forest for ASD classification. Their analysis revealed that images that contain multiple faces provide significant differences in visual attention between ASD and TD individuals. Wu, et al. [58] proposed two machine learning approaches based on synthetic saccade generation and image classification with similar performance in terms of accuracy and AUC. Tao and Shyu [59] proposed a combination of CNN and long short-term memory (LSTM) networks to classify ASD and TD individuals. Exploiting a similar architecture, Chen and Zhao [43] proposed a multimodal approach to utilise information from behavioural modalities captured during photo-taking and image-viewing tasks, resulting in higher performance in both modalities. Using an additional dataset that contains people looking at other people/objects in the scene, Fang, et al. [60] proposed a DNN that achieved a higher accuracy when compared to a previous model [33]. Rahman, et al. [61] used several saliency prediction models and compared the performance of SVM and XGBoost. Observing that not all images highlight significant differences in visual attention between ASD and TD participants, Xu, et al. [62] used structural similarity between ASD and TD saliency maps to identify a subset of images in which a new bio-inspired metric was applied to identify ASD participants. Wei, et al. [63] proposed a dynamic filter and spatiotemporal feature extraction for ASD diagnosis, achieving the highest accuracy and similar specificity and AUC scores when compared to previous models [56–59]. Liaqat, et al. [64] proposed two ML approaches that include a branched MLP approach and an image-based approach for ASD classification and found that the latter approach resulted in slightly better performance. Mazumdar, et al. [65] extracted different handcrafted and DL features and compared 23 ML algorithms to identify individuals with ASD. Their results were among the top 4 performing models across different metrics when compared to previous models [56,59,64].

## Eye-tracking on dynamic stimuli for ASD diagnosis

Prior research explored the possibility of using the eye-tracking paradigm during video-viewing tasks to identify specific neurological disorders. For example, Tseng, et al. [66] extracted low-level features from eye movement recorded from 15 minutes of videos and used an ML model to identify participants with attention deficit hyperactivity disorder, fetal alcohol spectrum disorder and Parkinson's disease. Although this work did not include ASD classification, it accentuates the efficacy of using eye-tracking on dynamic stimuli to identify the mental states of participants.

As shown in Table 4, there are recent works that utilise dynamic stimuli to differentiate ASD from TD subjects. Wan, et al. [67] investigated the difference in fixation times between ASD and TD children watching a 10-second video of a female speaking. Their results revealed that fixation times at the mouth and body could significantly discriminate ASD from TD with a classification accuracy of 85.1%. Jiang, et al. [68] collected eye-tracking data during a dynamic affect recognition evaluation task, extracted handcrafted features and used a random forest classifier to identify ASD individuals. Zhao, et al. [69] collected eye-tracking data during a live interaction with an interviewer, extracted handcrafted features and employed four ML classifiers to identify individuals with ASD. These prior studies rely on handcrafted features that may provide less discriminative information between TD and ASD individuals.

Numerous studies employed an image classification approach based on a published dataset that contains the visualisation of eye-tracking data (i.e., scanpath images) of the participants during the experiment [45]. For example, Carette, et al. [45,70] used the raw pixel values as features and compared ML and DL algorithms for ASD classification. Their results revealed that DL algorithms achieved the highest performance when compared to ML models. Elbattah, et al. [71] trained a deep autoencoder and used a k-means clustering approach on the learned

**Table 4. Eye tracking on dynamic stimuli for ASD diagnosis.**

| Authors | Mean age (SD) in years | Sample size | Stimuli | Input used | Method | Conclusions |
|---|---|---|---|---|---|---|
| Wan, et al. [67] | ASD: 4.60 (0.70) TD: 4.80 (0.40) | ASD: 37 TD: 37 | Dynamic, 10-second video of a female actor speaking | Eye-tracking data of the participant | They defined ten AOIs and computed different fixation time ratio. Afterwards, they used SVM to determine which AOI can be used for classification. | They found that using fixation times at the mouth and body results in an ASD classification accuracy of 85.1%, sensitivity of 86.5% and specificity of 83.8%. |
| Jiang, et al. [68] | ASD: 12.74 (2.45) TD: 14.11 (5.09) | ASD: 23 TD: 35 | Combination of static and dynamic stimuli | Dynamic stimuli with the associated eye-tracking data of the participant | They computed eye-tracking variables (e.g., response time, fixation locations, length, frequency, duration, saccadic amplitude) and extracted face features using a DL model. They then used RF for classification. | The combination of all the handcrafted and extracted features resulted in a classification accuracy of 72.5%. Using a soft voting approach, the classification accuracy increased to 86.2% in identifying ASD participants. |
| Zhao, et al. [69] | ASD: 8.30 (2.09) TD: 9.07 (2.25) | ASD: 19 TD: 20 | Dynamic, structured face-to-face conversation with a female interviewer | Dynamic stimuli with the associated eye-tracking data of the participant | They computed visual fixation ratios in four pre-defined AOIs across four sessions and added five features on session length, resulting in 21 features. Afterwards, they compared combinations of these features using different ML classifiers (e.g., SVM, LDA, DT and RF). | Their model that used the total session length, percentage of visual fixation time on the mouth AOI and the percentage of visual fixation time on the body as features achieved the highest classification accuracy. Looking at a single feature, the total session length was an effective discriminative feature. |
| Carette, et al. [45] | All participants: 7.88 (NR) | ASD: 29 TD: 30 | Combination of static and dynamic stimuli that depict naturalistic scenes, initiate joint attention and static face or objects. | They visualised the eye-tracking data of a participant as a scanpath image. Using the scanpath images, they converted it to a grayscale image and rescaled for further processing. | They defined the ASD classification as an image classification problem using a logistic regression model. | Their result achieved an AUC of 0.819 based on 10-fold cross validation. |
| Carette, et al. [70] | Same as Carette, et al. [45] | Same as Carette, et al. [45] | Same as Carette, et al. [45] | Same as Carette, et al. [45] | They defined the ASD classification as an image classification problem using several ML and ANN models. | Their MLP achieved the best performance when compared to ML models. They noted that there was no performance increase as the complexity is increased. |
| Elbattah, et al. [71] | Same as Carette, et al. [45] | Same as Carette, et al. [45] | Same as Carette, et al. [45] | Same as Carette, et al. [45] | They trained an autoencoder for feature extraction. Afterwards, they implemented a k-means clustering algorithm and analysed the cluster qualities in terms of ASD classification. | They showed that by using a clustering technique on the latent space representation in the autoencoder bottleneck, they could get a cluster that contains a high percentage of ASD participants, suggesting that the algorithm can be used for ASD classification. |
| Akter, et al. [72] | Same as Carette, et al. [45] | Same as Carette, et al. [45] | Same as Carette, et al. [45] | Same as Carette, et al. [45] | Using the scanpath images, they implemented a k-means clustering algorithm to divide the data into four groups. They trained different ML models in each cluster for classification. | Their results showed that the MLP achieved the best performance on different metrics when compared to ML models. |

*(Continued)*

**Table 4.** (*Continued*)

| Authors | Mean age (SD) in years | Sample size | Stimuli | Input used | Method | Conclusions |
|---|---|---|---|---|---|---|
| Cilia, et al. [73] | ASD: 7.58 (2.50) TD: 8.00 (2.67) | ASD: 29 TD: 30 | Same as Carette, et al. [45] | Scanpath images | They developed a four-layer CNN interspersed with pooling layers and a final FCLs for classification. | Their model achieved an accuracy of around 90%, sensitivity of around 83% and a precision of around 80%. |
| Kanhirakadavath and Chandran [74] | Same as Carette, et al. [45] | Same as Carette, et al. [45] | Same as Carette, et al. [45] | Same as Carette, et al. [45] | They compared two frameworks: (1) PCA for feature extraction and different ML techniques for classification. (2) CNN for feature extraction and different numbers of FCLs for classification. | Their results showed that the deep learning approach achieved higher performance across different metrics when compared to the different ML approaches. |
| Gaspar, et al. [75] | Same as Carette, et al. [45] | Same as Carette, et al. [45] | Same as Carette, et al. [45] | Scanpath images | Their approach is a kernel extreme learning machine that uses giza pyramids construction metaheuristic algorithm for kernel parameters optimisation. They compared this technique to other optimisation algorithms, as well as ML algorithms, in terms of classification accuracy. | Their proposed pipeline achieved the highest performance on different metrics when compared to other optimisation algorithms. In addition, their model achieved the highest performance on difference metrics when compared to other ML algorithms. |
| Ahmed, et al. [76] | Same as Carette, et al. [45] | Same as Carette, et al. [45] | Same as Carette, et al. [45] | Scanpath images | They developed three models that are based on ML, DL and hybrid techniques for classification. | The highest performing model was the ANN that uses the features extracted from the snake algorithm trained for image segmentation. |
| de Belen, et al. [14] | All participants: 4.60 (0.50) | ASD: 17 TD: 17 | Same as Pierce, et al. [11], Pierce, et al. [12] and Moore, et al. [13] | Dynamic stimuli with the associated eye-tracking data of the participant | They trained a VAM and used SVM for classification. | Using different number of fixations, their model achieved an accuracy of 68%-100%, sensitivity of 57%-100% and specificity of 65%-100%. |
| Oliveira, et al. [15] | Range: 3 to 18 | ASD: 76 TD: 30 | Dynamic, similar to GeoPref that contains biological and geometric movements | Dynamic stimuli with the associated eye-tracking data of the participant | For the entire video duration, they created two sets (one for each group) that contain the aggregated fixation locations on each frame. They created a group-specific fixation map which was then used to train VAMs. Afterwards, an individual classification was performed based on the VAMs. | Their model achieved an average precision of 90%, average recall of 69% and average specificity of 93%. |
| Fan, et al. [77] | All participants: Range: 3 to 13 | ASD: 21 TD: 47 | Point-light biological motion animation with upright/inverted persons that perform different actions. | They defined 5 'zones' where the visual attention of the participant is allocated. Afterwards, they computed data distribution within these zones. | They used the fixation distribution in different zones to identify zones helpful for classification. They trained an SVM for classification. | Their method achieved an AUC of 0.95. |

**Table 4.** (Continued)

| Authors | Mean age (SD) in years | Sample size | Stimuli | Input used | Method | Conclusions |
|---|---|---|---|---|---|---|
| Fang, et al. [78] | Age range: ASD: 4 to 10 TD: 2 to 15 | ASD: 33 TD: 50 | Same as Fan, et al. [77] | Same as Fan, et al. [77] | Using the extracted features, they compared kNN, Gaussian Naïve Bayes and Nonlinear SVM for ASD classification. | Their results showed that the nonlinear SVM achieved higher performance than the other MLP approaches. |
| Carette, et al. [79] | All participants: 8 to 10 | ASD: 17 TD: 15 | Dynamic, an actor initiating bids of joint attention | Eye-tracking data of the participants | Different saccadic movement variables were calculated as input to a two-layer LSTM network for classification. | Their model was able to identify ASD participants from TD participants with an accuracy of 83%. |
| Putra, et al. [80] | ASD: 5.00 (0.60) TD: 4.60 (0.40) | ASD: 21 TD: 31 | Dynamic, CatChicken game | Eye-tracking data of the participants | They extracted different features and used the AdaBoost metalearning algorithm. | Their approach achieved an accuracy of 88.6%. |

ANN: Artificial Neural Network, AOI: Area Of Interest, ASD: Autism Spectrum Disorder, AUC: Area Under the Curve, CNN: Convolutional Neural Networks, DL: Deep Learning, DT: Decision Tree, FCL: Fully-Connected Layer, kNN: k-Nearest Neighbour, LDA: Linear Discriminant Analysis, LSTM: Long Short-Term Memory, ML: Machine Learning, MLP: Multi-Layer Perceptron, NR: Not reported, PCA: Principal Component Analysis, RF: Random Forest, SD: Standard deviation, SVM: Support Vector Machine, TD: Typically Developing, VAM: Visual Attention Model

latent representation to identify clusters of participants. Their analysis revealed that an identified cluster contained a high percentage of ASD participants, suggesting that the algorithm can be used for ASD classification. Using a similar unsupervised learning approach, Akter, et al. [72] performed k-means clustering to divide the dataset into 4 groups and compared different ML models to identify participants with ASD. Cilia, et al. [73] used CNN and a fully-connected layer to predict ASD participants. Similarly, Kanhirakadavath and Chandran [74] compared Principal Component Analysis (PCA) and CNN for feature extraction and different ML and DL models for ASD classification. Gaspar, et al. [75] performed additional image augmentation to generate more training data. Afterwards, they used a kernel extreme learning machine optimised using the Giza Pyramids Construction metaheuristic algorithm to identify ASD individuals. Their approach achieved higher performance when compared to ML approaches. Ahmed, et al. [76] compared ML, DL and a combination of both approaches for ASD diagnosis. The results in these prior studies suggest that DL models for feature extraction and ASD classification perform better when compared to traditional ML approaches.

There are also prior studies that explored the use of dynamic stimuli that are effective in evoking significant differences in visual attention of ASD and TD participants. For example, de Belen, et al. [14] used the GeoPref Test [11,12] in EyeXplain Autism, a system for eye-tracking data analysis, automated ASD prediction and interpretation of deep learning network predictions. Recently, Oliveira, et al. [15] used similar video stimuli, trained a visual attention model and utilised an ML model to identify individuals with ASD. Fan, et al. [77] and Fang, et al. [78] used biological motion stimuli and different ML classifiers for ASD diagnosis. Using a stimulus for initiating joint attention, Carette, et al. [79] extracted features related to saccadic movement (e.g., amplitude, velocity, acceleration) and trained an LSTM network to predict three diagnostic groups (i.e., ASD, TD, unclassified). Putra, et al. [80] collected eye-tracking data during Go/No-Go tasks, identified spatial and auto-regressive temporal gaze-related features that differ significantly between ASD and TD participants and applied an AdaBoost meta-learning algorithm to identify participants with ASD.

Although previous studies utilised dynamic stimuli, the most common approach was to convert the participant's eye-tracking data into an image, potentially losing spatiotemporal information that can be leveraged for classification. In addition, this approach disregards the pixel information around the fixation, a crucial insight into what part of the stimuli attracts human attention. In this paper, we propose a DNN approach that utilises dynamic saliency prediction to identify individuals with ASD.

While previous works have investigated the feasibility of leveraging visual attention in identifying individuals with ASD, limited research has been conducted to explore the effectiveness of exploiting the dynamic visual attention of the participant in ASD classification. Our approach utilises eye-tracking data captured during a dynamic stimulus viewing task. Our approach follows a similar deep learning framework reported in the literature [33], however it provides an extension from static stimuli, widening the diagnostic paradigm to include dynamic stimuli.

## Eye-tracking in ASD risk and symptom severity prediction

Although there has been a great deal of research on the use of eye-tracking in ASD diagnosis, relatively little research focus on other applications, such as automatically predicting the risk of ASD (e.g., low, medium and high) and symptom severity, as shown in Table 5. Nevertheless, previous studies provide insights into the potential use of eye tracking in symptom severity prediction. For example, Kou, et al. [81] found that a reduction in visual preference for social

**Table 5. Eye tracking in ASD risk and symptom severity prediction.**

| Authors | Mean age (SD) in years | Sample size | Stimuli | Input used | Method | Conclusions |
|---|---|---|---|---|---|---|
| Canavan, et al. [23] and Fabiano, et al. [24] | Two experiments: Experiment 1: Range: between 2 and 60 years old Experiment 2: Range: between 2 and 40 years old | Two experiments: Experiment 1: 257 with different risk types (low, medium, high and confirmed ASD) Experiment 2: 237 (subset of the first experiment) | Image and Video | They used the raw eye-tracking data (x and y locations), handcrafted features (e.g., average fixation duration, velocity), age and gender | They compared different ML and DL algorithms for ASD risk prediction. | Their approach achieved a maximum classification rate of 93.45%. |
| Revers, et al. [16] | Range: between 3 and 16 years old. | NSG: 49 SG: 39 | Same as Pierce, et al. [11], Pierce, et al. [12] and Moore, et al. [13] | They used the stimulus and the associated eye-tracking data of the participant. | They trained two computational models [83] to generate saliency maps of SG and NSG. Afterwards, they used RELIEFF algorithm to select features for classification [84]. | Their model achieved an average accuracy of 88%, precision of 70%, sensitivity of 87% and specificity of 60% for ASD symptom severity prediction. |
| Carette, et al. [70] | Same as Carette, et al. [45] | Same as Carette, et al. [45] | Same as Carette, et al. [45] | Same as Carette, et al. [45] | They defined the symptom severity prediction as an image classification problem using ANN models. | Their model achieved an average accuracy of around 83%. Their model was able to better identify TD participants compared to other ASD symptom severity. The prediction accuracy of symptom severity labels was 20% lower and worse for severe ASD participants. |

ANN: Artificial Neural Network, ASD: Autism Spectrum Disorder, ML: Machine Learning, NSG: Non-Severe Group, SD: Standard Deviation, SG: Severe Group, TD: Typically Developing

scenes is significantly correlated with the ADOS social affect score, which may be useful in severity prediction. On the other hand, Bacon, et al. [82] found that a higher visual preference of toddlers for geometric scenes is significantly correlated with later symptom severity at school age, further suggesting the clinical utility of eye tracking for ASD symptom severity prediction.

Recently, Revers, et al. [16] trained two computational models [83] to generate saliency maps of severe and non-severe groups and used the RELIEFF algorithm [84] to select the most important features for classification. Afterwards, a neural network was trained to identify symptom severity for each fixation made by the participant. The final prediction is considered to be severe if more than 20 fixations were classified as severe by the trained neural network. Their approach obtained an average accuracy of 88%, precision of 70%, sensitivity of 87% and specificity of 60% in predicting symptom severity.

In a slightly different problem, Canavan, et al. [23] and Fabiano, et al. [24] proposed a method for predicting ASD risk using eye gaze and demographic feature descriptors (e.g., age and gender). Handcrafted features, such as average fixation duration and average velocity, were tested on four different classifiers, namely random forests, decision trees, partial decision trees and a deep forward neural network. Although their results with a maximum classification rate of 93.45% are promising, it is crucial to compare their handcrafted features to features learned by modern deep learning models and determine if the latter improves the risk prediction accuracy. In this paper, we present the same DNN approach we used in ASD classification to predict the level of ASD-related symptoms.

## Materials and methods

In this work, we used a data-driven approach to extract rich features learned from a dynamic stimulus to identify participants with autism and predict the level of ASD-related symptoms. In Fig 1, an overview of the proposed approach is provided. The method is divided into different stages, including eye-tracking data collection, dynamic saliency detection trained on the

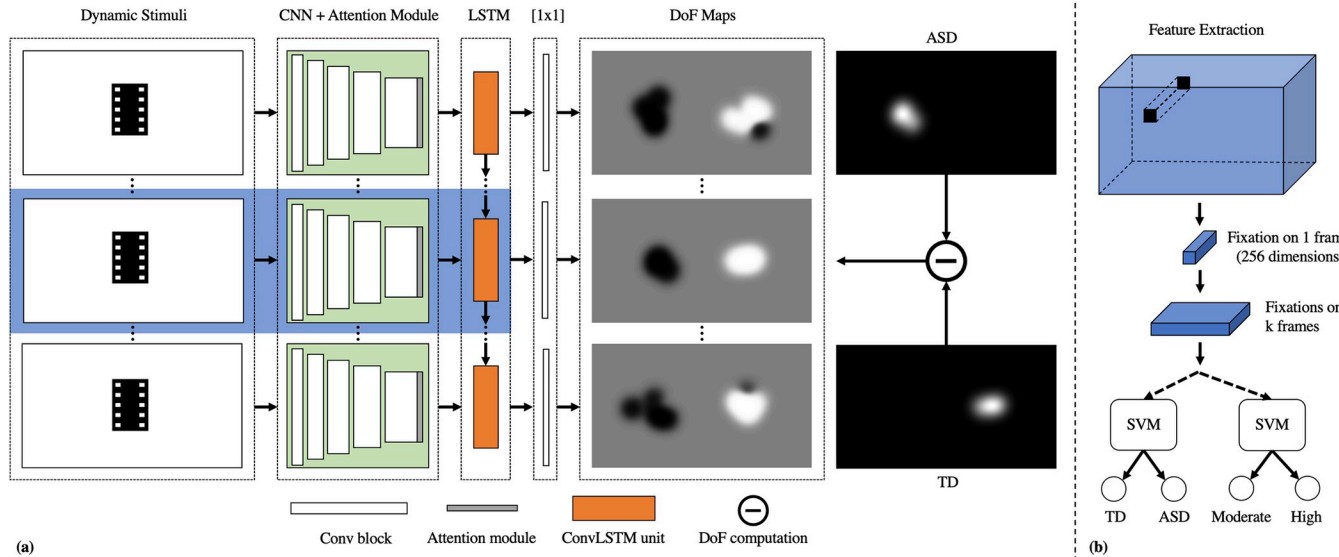

**Fig 1. Overview of the proposed feature learning/extraction, classification and symptom severity prediction approach.** (a) Given a video input, per-frame features are learned using an end-to-end approach to predict the difference of fixation (DoF) maps. (b) Extracted features at fixated pixels from each fixation stage are cascaded and passed on to an SVM to identify individuals with ASD and predict the level of ASD-related symptoms.

difference of fixations between ASD and TD individuals, and SVM-based classification and severity prediction. This study was approved by the Human Research Ethics Committee of the University of New South Wales. Written informed consent was obtained from the parents/legally authorised representatives of the participants. All methods were carried out in accordance with relevant guidelines and regulations.

## Eye-tracking

**Participants.** There were 57 children (9 females) in the ASD group and 17 children (9 females) in the TD group. Participants were matched by their age at the time of the study. 24 children in the ASD group were recruited from an Autism Specific Early Learning and Care Centre (ASELCC) and 33 children were recruited from the Child Development Unit (CDU) of a Children's Hospital. The TD children were recruited from a children's services preschool. All participants in the ASD group met the criteria for ASD based on the Diagnostic and Statistical Manual of Mental Disorders (DSM-5) [85] criteria and the diagnosis of ASD was confirmed using the Autism Diagnostic Observation Schedule (ADOS), Second Edition [86]. Of the 57 ASD children, there were 24 who showed high ASD-related symptoms and 33 had moderate symptoms. There are no specific exclusion criteria for the ASD group in this study. The TD group's exclusion criteria included known neurodevelopmental disorders, significant developmental delays and known visual/hearing impairments. No child had any visual acuity problems.

**Dynamic stimulus.** We used the GeoPref Test [11,12] dynamic stimulus, which has been shown to be an effective stimulus for detecting ASD subgroups. This stimulus consists of dynamic geometric images (DGIs) on one side and dynamic social images (DSIs) on the other. The DGIs were constructed from recordings of animated screen-saver programs. The DSIs were produced from a series of short sequences of children performing yoga exercises. It included images of children performing a wide range of movements (e.g., waving arms and appearing as if dancing). The stimulus contained a total of 28 different scenes and was presented in order, based on the originally published stimulus [11,12]. It has a resolution of 1281 x 720 pixels and contains a total of 1,488 frames, which is equivalent to 61 seconds of video playback.

**Eye-tracking apparatus and procedure.** Participants were tested using the Tobii X2-60 eye tracker and eye-tracking data was processed using Tobii Studio software to identify fixations and saccades. Eye movements were recorded at 60 Hz (with an accuracy of 0.5°) during the dynamic stimuli viewing. Each participant was seated approximately 60 cm in front of a 22" monitor with a video resolution of 1680 x 1050 pixels in a quiet room and shown dynamic visual stimuli in full-screen. A built-in five-point calibration in Tobii studio was completed before administering the task for accurate eye gaze tracking. The calibration procedure required gaze following on an image of an animal paired with auditory cues, starting with the centre of the screen, and moving across the four corners of the screen. The eye-tracking procedure was conducted during a clinical assessment or the intake assessment for entry to an early intervention program. Multiple attempts were made to ensure that the eye tracker had been calibrated properly for accurate data collection. Multiple attempts were also made to ensure that the participants were engaged during the experiment. As a result, depending on the capacity of the child, the procedure was conducted over 2 to 3 sittings or with smaller breaks in between. The overall clinical assessment and eye-tracking procedure were completed in approximately 2.5h per participant.

**Data processing and statistical analysis.** Tobii Studio's I-VT filter [87] was used to process the raw eye-tracking data, exclude random noise and define fixations for further analysis.

More specifically, short fixations (<100ms) were discarded and adjacent fixations (75ms, 0.5˚) were merged. Trials were excluded if the total fixation duration was less than 15 seconds. That is, to be included, the participant should be looking at the stimulus for approximately 25% of the entire video duration. Once included, the eye-tracking data captured during the entire length of the stimulus are used for training and evaluation.

A calibration quality assessment was performed to rule out the possibility of eye-tracking data quality as a confounding factor. In this assessment, a toy accompanied by a sound was used to attract the participants' gaze to the calibration point in the middle of the screen. The mean distance between the detected fixation locations and the calibration point was calculated as a measure of accuracy. A t-test showed no significant difference between the groups, suggesting that data quality did not differ between the two groups: t(64) = -0.445, p = .658, ASD: 45.89 pixels (22.67), TD: 48.76 pixels (19.00).

An additional data quality assessment was performed to determine the overall nature of the visual attention of the participants to the stimuli. A t-test showed no significant difference in visual attention between groups: t(72) = 0.011, p = .991, ASD: 37.13 seconds (12.03), TD: 37.10 seconds (8.07). These analyses of quality suggest that it is unlikely that differences in data quality and general visual attention influenced the results.

An independent-samples t-test was used to investigate differences in visual attention across two groups for diagnosis (ASD vs. TD) and severity prediction (moderate vs. severe). All statistical analysis was performed in IBM SPSS Statistics Version 26.

## Computation of per-frame saliency maps

Saliency detection models are typically optimised to detect salient features in a scene. They are trained on a probability distribution of eye fixations, called the fixation map. The per-frame fixation maps of each participant group were generated from the eye movement data collected in the study. For a given frame, all fixation points of the children in each group were overlaid in a binary map, in which the fixation points were set to 1 on a black background (value set to 0). The resulting per-frame fixation maps were smoothed with a Gaussian kernel (bandwidth = 1˚) and normalised by the sum to generate per-frame visual attention heatmaps (labelled ASD and TD heatmaps in Fig 2).

## Computation of per-frame difference of fixation (DoF) maps

Similar to Jiang and Zhao [33], our network was optimised on the difference of fixation (DoF) maps, highlighting the difference in visual attention between TD and ASD individuals. Since our approach uses a dynamic stimulus, we predict DoF maps on each frame. In particular, let $I^+$ and $I^-$ be the fixation maps for the ASD and TD groups, respectively. The DoF map of a frame is computed as:

$$D = \frac{1}{1 + e^{-I/\sigma_I}}$$

where $I = I^- - I^+$ is a pixel-wise subtraction of fixation maps and $\sigma_I$ represents the standard deviation of I.

The resulting DoF maps highlight the difference in visual attention between ASD and TD individuals (refer to Fig 2). The white regions of the DoF map illustrate the visual attention of TD individuals while the black regions are for ASD individuals. Note that this is the opposite of the DoF computation elsewhere [33]. This also resulted in better training performance compared to DoF maps that highlight more fixations of the ASD group.

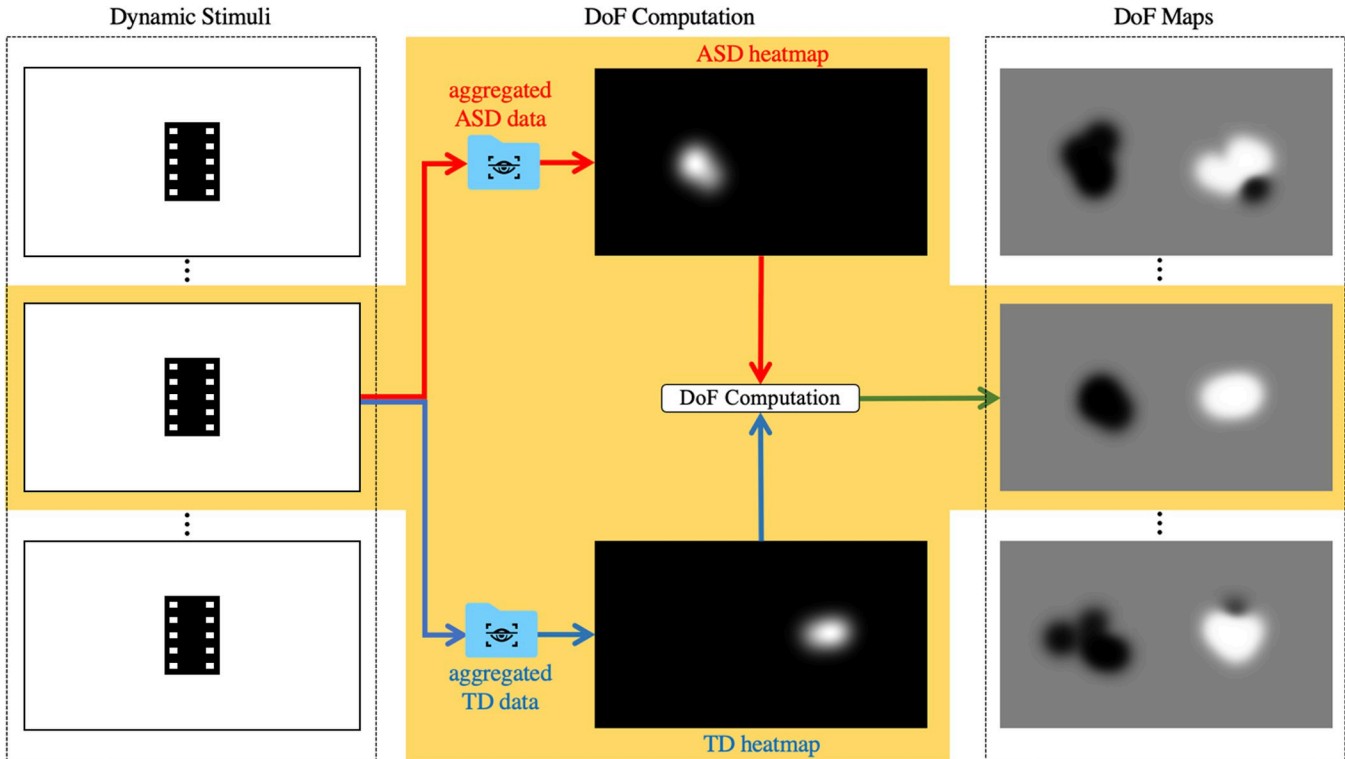

**Fig 2. Overview of the computation of difference of fixation (DoF) maps.** On the left, the dynamic stimulus is analysed per frame. In the middle, the eye-tracking data in each participant group are aggregated and the difference is computed for each frame. On the right, the TD heatmaps are in white, while the ASD heatmaps are in black.

## Per-frame prediction of difference of fixation maps

As shown in Fig 3, ACLNet [88], one of the best models available for dynamic saliency detection, is used for feature extraction. It consists of a CNN-LSTM network with an attention mechanism to enable fast, end-to-end saliency prediction. Since ACLNet already contains an attention network trained on TD individuals, we trained and fine-tuned our model with DoF maps that highlight more fixations of the TD group.

Our model was optimised using the following loss function [89] which considers three different saliency evaluation metrics instead of the binary-cross entropy loss used before [33]. We denote the predicted difference of fixation map as $Y \in [0,1]^{28 \times 28}$ and the ground truth saliency map as $Q \in [0,1]^{28 \times 28}$. Our loss function combines Kullblack-Leibler (KL) divergence, the Linear Correlation Coefficient (LCC) and the Normalised Scanpath Saliency (NSS) similar to prior work [88]:

$$L = L_{KL} + 0.1 L_{LCC} + 0.1 L_{NSS}$$

$L_{KL}$ is widely used for training saliency models and is computed by:

$$L_{KL}(Y, Q) = \sum_x Q(x) \log \frac{Q(x)}{Y(x)}$$

                                                  

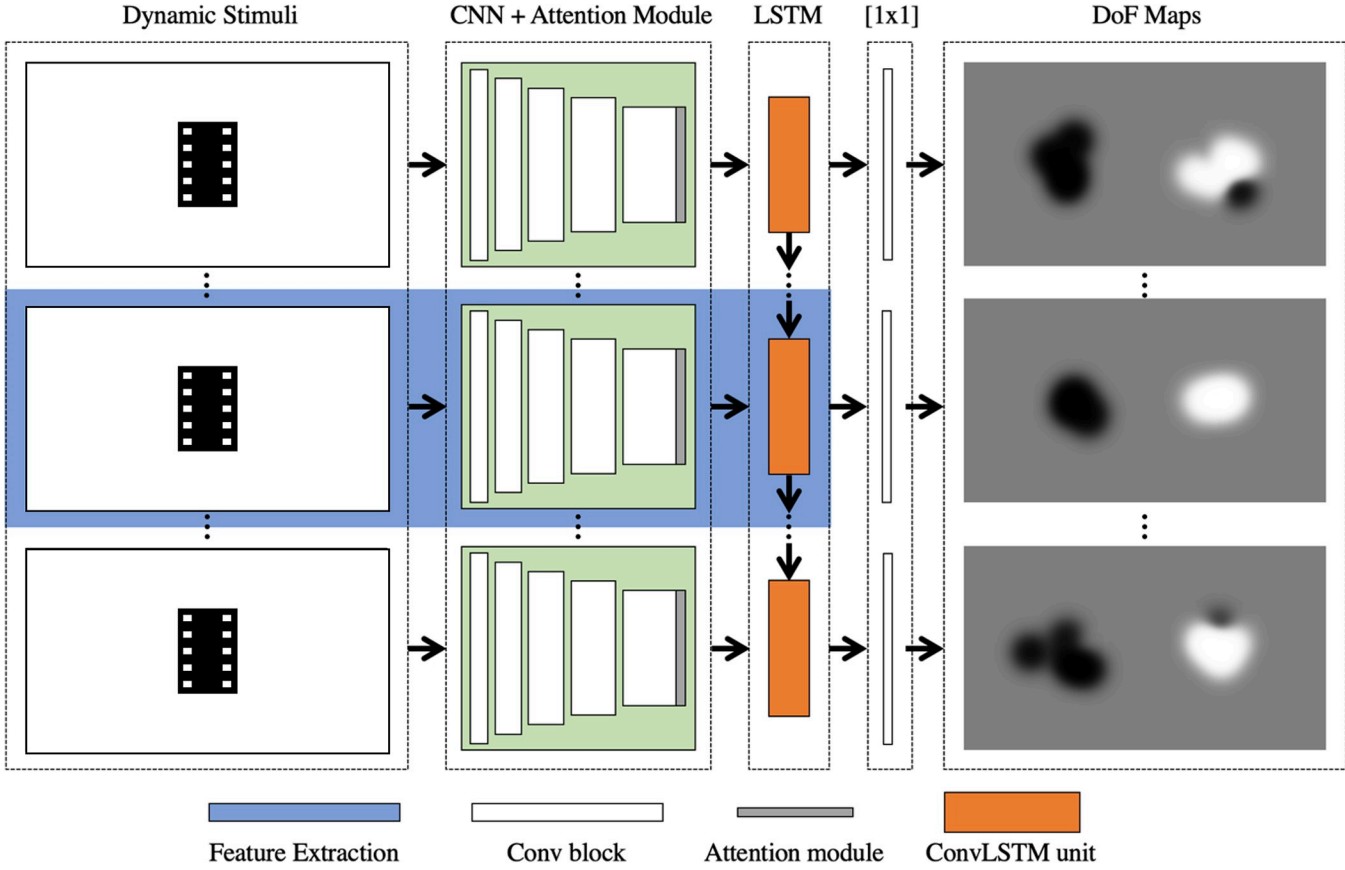

**Fig 3. Overview of the approach for learning the difference of fixation (DoF) maps.**

$L_{LCC}$ measures the linear relationship between Y and Q:

$$L_{LCC}(Y, Q) = -\frac{cov(Y, Q)}{\sigma(Y)\sigma(Q)}$$

where $cov(Y, Q)$ is the covariance of Y and Q while $\sigma$ is the standard deviation.

$L_{NSS}$ is defined as:

$$L_{NSS}(Y, Q) = -\frac{1}{N}\sum_{x}\bar{Y}(x) \times Q(x)$$

where $\bar{Y} = \frac{Y-\mu(Y)}{\sigma(Y)}$ and $N = \Sigma_x Q(x)$. It computes the mean of scores from the normalised saliency map $\bar{Y}$ at the predicted DoF maps Y.

## Training protocol

Our classification and severity prediction models are iteratively trained with sequential DoF maps and image data. We train the model by using a loss defined over the predicted dynamic saliency maps from convLSTM. Let $\{Y_t^d\}_{t=1}^T$ and $\{Q_t^d\}_{t=1}^T$ denote the predicted dynamic

saliency maps and continuous difference of fixation maps. We minimise the following loss:

$$L^d = \sum_{t=1}^{T} L(Y_t^d, Q_t^d)$$

The parameters of ACLNet are initialised to the pre-trained parameters [88]. The network is then fine-tuned on the current dataset.

## ASD classification and symptom severity prediction

Once the model has been trained to predict DoF maps of ASD and TD individuals from a given dynamic stimulus, feature extraction and classification are performed, with Fig 4 illustrating the process [14]. Based on the eye-tracking data, we determined the fixation positions and the corresponding frames in which they were recorded. Each saccade-fixation pair was considered a fixation stage. For each fixation stage, features were extracted from the corresponding fixation position on the feature map obtained from the convLSTM output (note that the convLSTM output is upsampled 4 times before extracting the feature map). More specifically, given a frame where a fixation has been identified, the feature map at the corresponding fixation is extracted, which results in a 256-dimensional feature vector at each fixation. For a corresponding number of fixation stages, feature vectors for all fixations are concatenated in their temporal order starting from the first fixation to the last fixation stage. This serves as the feature space in which classification is performed. If there were fewer identified fixations, zeros are appended at the end. We explored the number of fixation stages that provided the best performance.

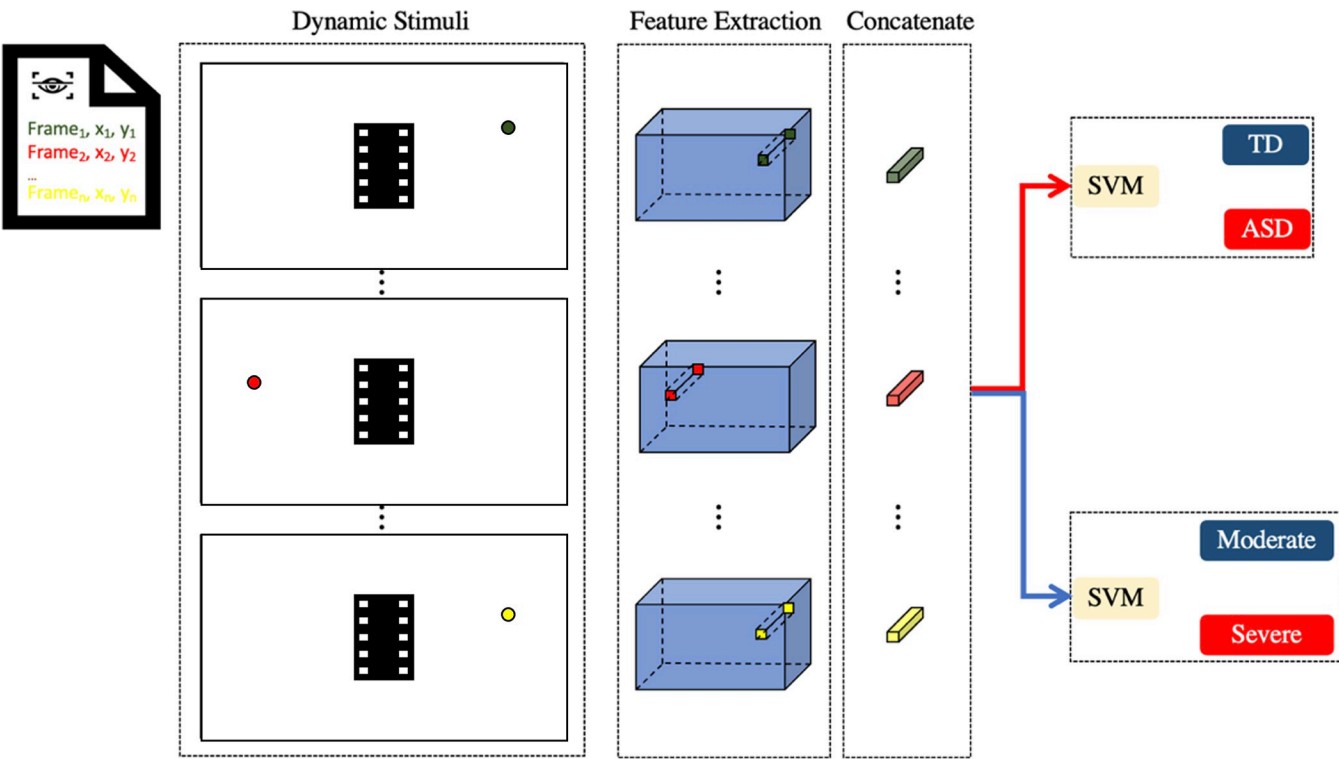

**Fig 4. Overview of the approach for feature extraction and classification.**

A linear decision boundary between ASD and TD individuals was determined by training an SVM on the extracted features. In addition, another SVM model was trained on the DoF maps of moderate and high ASD individuals to predict autism severity. We used the ADOS-2 calibrated severity scores (CSS) as ground truth to determine the ASD severity. Participants with ADOS CSS of 5–7 are considered to have moderate symptoms, while those with ADOS CSS of 8–10 are considered to have more severe (high) symptoms.

## Experimental setup

**Training and testing protocols.**    We report the model's performance on ASD classification and symptom severity prediction using leave-one-out cross-validation (LOOCV). Given the unbalanced nature and the limited number of samples in the dataset, LOOCV is used to provide an almost unbiased estimate of the probability of error [90]. In addition, it allows us to maximise the number of training samples per fold unlike in a k-fold validation approach. While a stratified k-fold cross-validation strategy may account for the group imbalance that is present in our dataset, it results in smaller training samples per fold. However, removing a single sample from the training set done in LOOCV also does not drastically change the class distribution. The combination of being able to use as much training data as possible while also maintaining similar class distribution was the reason why we used LOOCV. The same evaluation approach has been employed in prior studies [14,33,34,43,68,69] in this application area.

**Implementation details.**    We implemented our model in Tensorflow with Keras and Scikit-learn libraries. During the training phase, we fine-tuned the network with Adam optimizer and a batch size of one image for a total of 20 epochs. The learning rate was set to 0.0001. We did not perform any dropout and data augmentation. L2 regularisation with the penalty parameter $C = 1$ was used for SVM classification.

**Evaluation metrics.**    We report on the performance of our model in terms of accuracy, sensitivity (i.e., true positive rate) and specificity (i.e., true negative rate) recorded at different numbers of fixations. Once the best number of fixations to be included in the classification was identified, the area under the receiver operating characteristic (ROC) curve and the confusion matrix were also computed. To obtain a meaningful area under the ROC curve (AUC) in an LOOCV, the output probability of the SVM for each fold (each consisting of just one subject) was saved and the AUC was computed on the set of these probability estimates. The computation of the confusion matrix was performed similarly using the predicted class to compare with the ground truth label.

**Computational load.**    The entire training procedure for each video stimulus takes about 1 hour with two NVIDIA 2080 Super and a 3.5GHz Intel processor (i7-7800X CPU). Once the model has been trained, feature extraction and SVM classification can be performed in less than 1 minute.

## Results

### Datasets

Children with ASD had a mean age of 4.63(standard deviation (SD) = 0.80) years and TD participants also had a mean age of 4.61 (SD = 0.47) years. There was no significant difference in age between the ASD and TD groups, $t(72) = 0.009$, $p = 0.993$.

### Eye-tracking data analysis

**ASD classification.**    It was previously shown that ASD individuals with severe symptoms tend to fixate more on the geometric stimuli than the social stimuli [11,12]. Shown in Fig 5 are

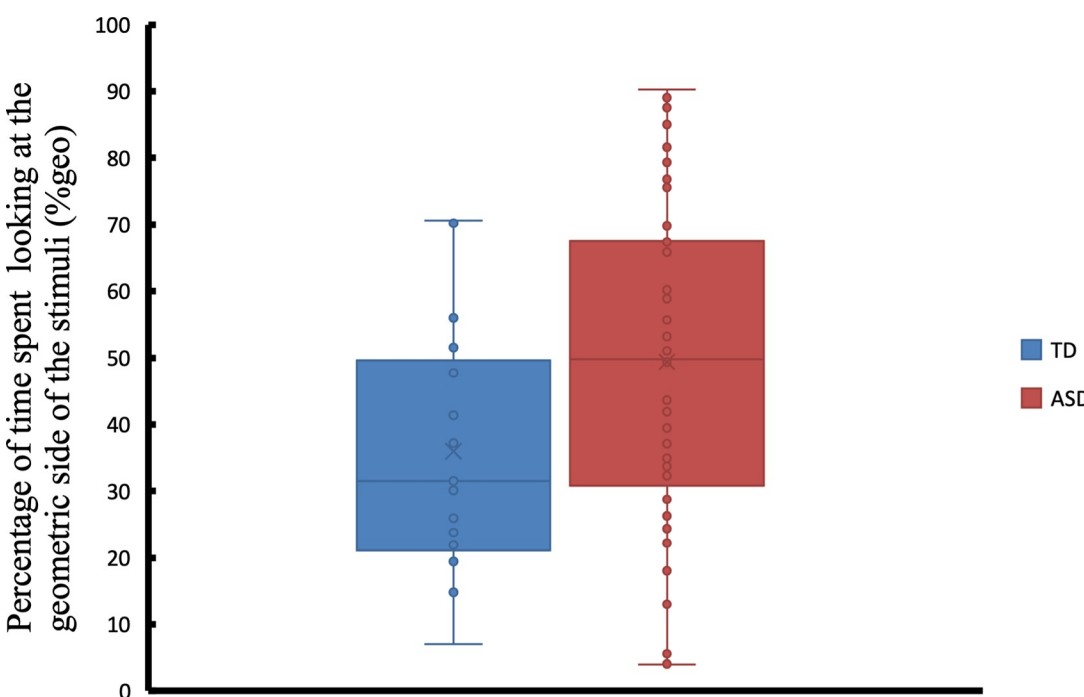

**Fig 5. Comparison of the percentage of time spent looking at the dynamic geometric stimuli (%geo) between TD and ASD participants.** Each box plot contains the interquartile range, the x marker corresponds to the mean value and the horizontal line inside corresponds to the median. Each sample is also visualised using dot points.

the %Geo values, the percentage of time spent looking at the dynamic geometric stimuli. % Geo values are computed by dividing the total fixation duration on the geometric stimuli by the total fixation duration on both geometric and social stimuli. Independent-samples t-test was used to compare %Geo for each diagnostic group. Similar to published results elsewhere [11–13], ASD participants in our study were significantly more attracted to dynamic geometric images when compared to TD participants (t = 2.11, p < .0386). On average, the ASD group spent 49.37% (standard deviation (SD) = 24.14%) of their attention looking at the dynamic geometric images, while the TD group spent 35.97% (SD = 18.58%) of their attention.

**ASD symptom severity prediction.** Shown in Fig 6 are the %Geo values, the percentage of time spent on looking at the dynamic geometric stimuli. There was no significant difference in the %Geo values between the moderate and severe ASD participants (t = 0.424, p < .6729). On average, ASD participants with moderate symptoms fixated around 48.21% (SD = 23.82%) of their attention on the geometric stimuli. On the other hand, ASD participants with severe symptoms spent 50.98% (SD = 25.00%) of their attention looking at the geometric stimuli. We also performed pair-wise comparisons between the TD participants and the two ASD partici-pant groups (i.e., moderate and severe). There was a significant difference in the %Geo values between ASD participants with severe symptoms and TD participants (t = 2.096, p < .0426). On the other hand, there was only a trend toward a significant difference in the %Geo values between ASD participants with mild symptoms and TD participants (t = 1.846, p < .0710).

In recent years, it has been shown that stimuli that have both dynamic geometric and social images can reliably separate the visual attention of ASD and TD individuals. We contribute to the literature by showing that a DNN-based approach using dynamic stimuli can result in highly accurate ASD classification and even predict the level of ASD-related symptoms with promising performance.

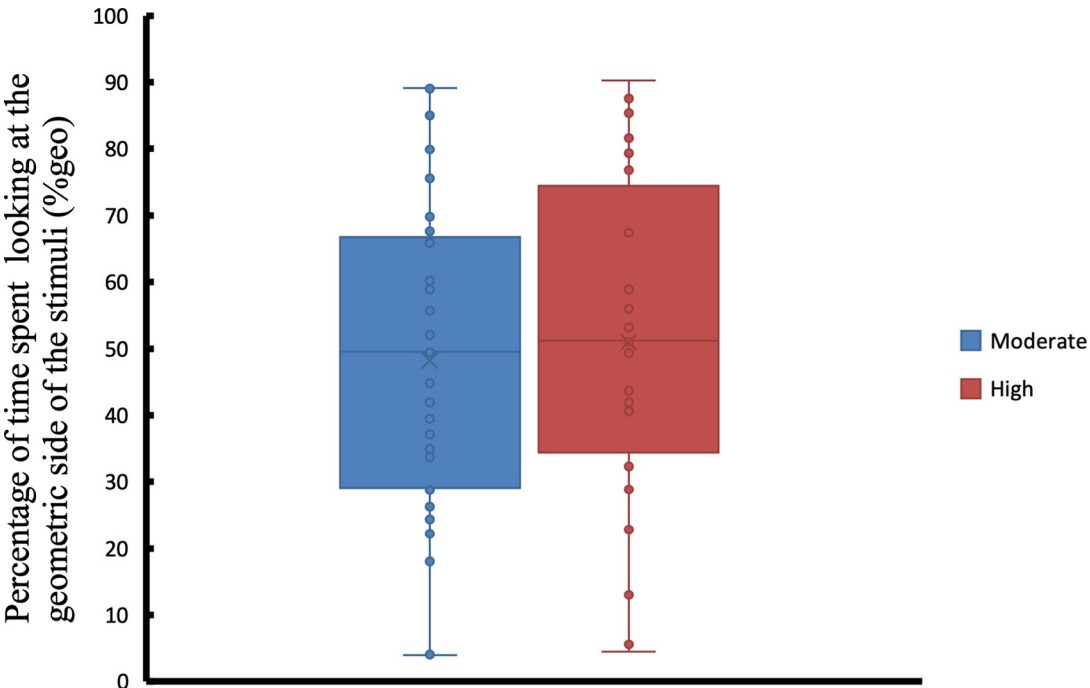

**Fig 6. Comparison of the percentage of time spent looking at the dynamic geometric stimuli (%geo) ASD participants with moderate and severe symptoms.** Each box plot contains the interquartile range, the x marker corresponds to the mean value and the horizontal line inside corresponds to the median. Each sample is also visualised using dot points.

**ASD classification performance.** In Fig 7, different performance metrics for ASD prediction on the GeoPref Test dynamic stimulus are shown. In Fig 7A, the accuracy, sensitivity and specificity of the model as the number of fixations (i.e., fixation length) increases are displayed. It can be observed that all measures generally increase as the number of fixations increases. In Fig 7B and 7C, the receiver operating characteristics (ROC) curve and the confusion matrix of the model that reported the highest accuracy (i.e., using the optimal fixation length) in Fig 7A are shown. The area under the ROC curve (AUC) of our model is 0.96, significantly higher than chance-level performance (AUC = 0.5). Our model achieved the highest accuracy of

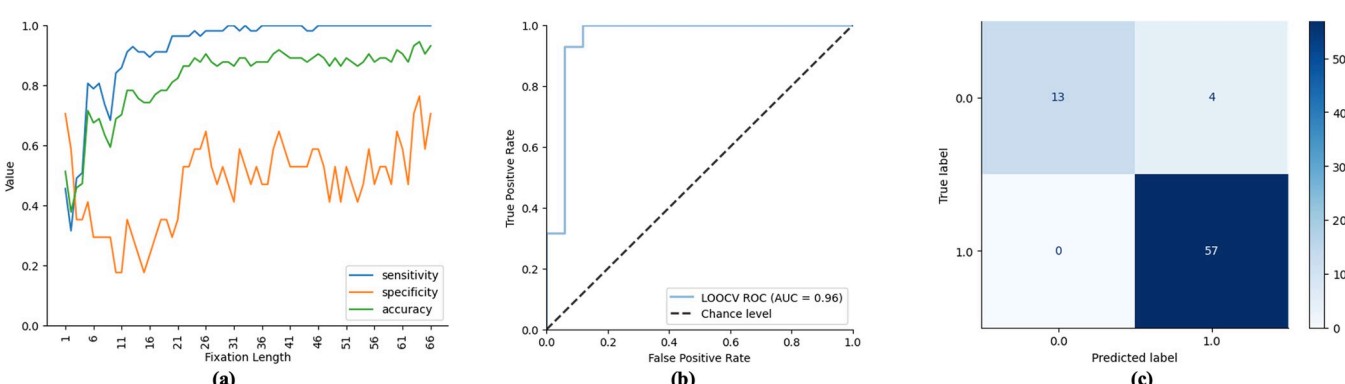

**Fig 7. Different performance metrics for ASD prediction.** (a) the plot of the model's sensitivity, specificity and accuracy as the number of fixations (i.e., fixation length) increases. (b) the plot of the area under the receiving operating curve of the best-performing model. (c) the confusion matrix of the best-performing model.

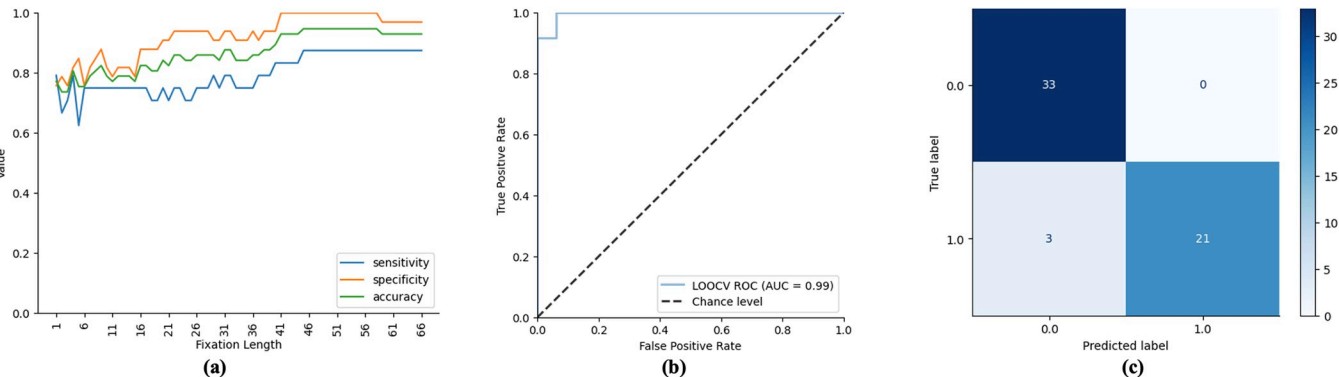

**Fig 8. Different performance metrics for ASD symptom severity prediction.** (a) the plot of the model's sensitivity, specificity and accuracy as the number of fixations (i.e., fixation length) increases. (b) the plot of the area under the receiving operating curve of the best-performing model. (c) the confusion matrix of the best-performing model.

94.59% when 64 fixations were included in the analysis. The high sensitivity of our model (highest value = 100%) suggests that it can reliably identify ASD children. On the other hand, the specificity of our model (highest value = 76.47%) suggests that it can reliably identify children without the disorder. Overall, four (4) children were mistakenly flagged as having the disorder despite not having it.

**ASD severity prediction performance.** Similar to the results of the diagnosis prediction, it can be observed in Fig 8A that all performance measures for ASD severity prediction generally increase as the number of fixations (i.e., fixation length) increases. In Fig 8B and 8C, the ROC curve and the confusion matrix of the model that reported the highest accuracy in Fig 8A are shown. Our model achieved the highest accuracy of 94.74% when 44 fixations were included in the analysis. The area under the ROC curve (AUC) of our model is 0.99, significantly higher than chance-level performance (AUC = 0.5). The high specificity of our model (highest value = 100%) suggests that it can reliably identify children with mild ASD. On the other hand, the high sensitivity of our model (highest value = 87.50%) suggests that it can reliably identify children with severe symptoms. Overall, three (3) children were mistakenly flagged as having severe diagnoses despite having milder symptoms.

## Comparison with other approaches

As outlined in the related work section, a straightforward comparison with previous approaches that utilise dynamic stimuli is not possible because the published dataset contains a visualisation of eye-tracking participants (i.e., scanpath images) rather than the stimuli used and the associated eye-tracking data that our model requires. Nevertheless, we compared our proposed approach with a simple thresholding method [11–13] and ML algorithms using handcrafted features [23,24].

**ASD classification.** Following the cut-off of %Geo > 69% to determine ASD participants in a similar study [11–13], we obtained a sensitivity of 22.80%, specificity of 88.23% and accuracy of 37.84%. The AUC obtained was 0.67. In comparison, our proposed model resulted in 77.2% higher sensitivity, 11.76% lower specificity and 56.75% higher accuracy when compared to solely utilising the %Geo values. Handcrafted features that include raw eye gaze points (x and y locations), average fixation duration, age and gender, were also used as input to a random forest regressor and a decision tree classifier for ASD classification similar to a previous study [23,24]. The random forest regressor achieved an accuracy of 72.97%, a sensitivity of

**Table 6. ASD classification results comparison with prior approaches.**

| Approach | Accuracy | Sensitivity | Specificity |
|---|---|---|---|
| Thresholding approach [11–13] | 37.84% | 22.80% | **88.23%** |
| Random forest regressor | 72.97% | 91.22% | 0.00% |
| Decision tree classifier | 58.11% | 70.18% | 17.65% |
| Ours | **94.59%** | **100%** | 76.47% |

91.22% and a specificity of 0%. On the other hand, the decision tree classifier achieved an accuracy of 58.11%, a sensitivity of 70.18% and a specificity of 17.65%.

Overall, our proposed model achieved the highest accuracy of 94.59%, the highest sensitivity of 100% and the second-best specificity of 76.47%. The comparison results in ASD classification suggest that our model better identified participants with ASD than the previous approaches, as shown in Table 6.

**ASD symptom severity prediction.** We also used the same cut-off of %Geo > 69% [11–13] to identify ASD participants with severe symptoms and obtained a sensitivity of 25.00%, specificity of 78.79% and accuracy of 43.24%. The AUC obtained was 0.54. Again, our proposed method showed promising results for severity prediction, resulting in a 62.50% increase in sensitivity, a 21.21% increase in specificity and a 51.5% increase in accuracy when compared to solely utilising the %Geo values. In comparison to our model, using handcrafted features and ML classifiers resulted in the same accuracy of 94.74%, slightly higher sensitivity of 91.67% and slightly lower specificity of 96.97%. Overall, our proposed model achieved the highest accuracy of 94.47%, the second-best sensitivity of 87.50% and the highest specificity of 100%. The comparison results in ASD symptom severity prediction suggest that our model better identifies participants with moderate symptoms than the previous approaches, as shown in Table 7.

## Discussion

Over the past decade, eye-tracking studies have revealed significant differences in visual attention between ASD and TD individuals. This motivated researchers to leverage recent advances in saliency prediction when designing a more quantitative approach to ASD diagnosis, as well as risk and symptom severity prediction. In this context, researchers have explored the use of static and dynamic stimuli during free-viewing tasks. The most common approach in the literature comprised of a traditional two-stage method that consists of a feature extraction stage followed by a classification stage. Increasing evidence suggests that the DL-based approach produced more discriminative features when compared to ML-based approaches. Classification methods that utilise DL also resulted in better performance than ML models. The rapid advances in DL approaches and the increasing number of publicly available datasets may help further advance the literature and improve classification performance. In this paper, we utilised a combination of DL and ML approaches for ASD diagnosis and symptom severity prediction.

**Table 7. ASD symptom severity prediction results comparison with prior approaches.**

| Approach | Accuracy | Sensitivity | Specificity |
|---|---|---|---|
| Thresholding approach [11–13] | 43.24% | 25.00% | 78.79% |
| Random forest regressor | **94.74%** | **91.67%** | 96.97% |
| Decision tree classifier | **94.74%** | **91.67%** | 96.97% |
| Ours | **94.74%** | 87.50% | **100%** |

Unlike prior research that utilised dynamic stimuli and converted the participant's eye-tracking data into an image for classification, we propose a data-driven approach utilising a dynamic saliency model to extract discriminative features from the stimuli and an ML approach based on eye-tracking data to automatically identify individuals with ASD. In addition, we show that the same approach can predict the level of ASD-related symptoms in preschool children. Our approach to identifying children with ASD offers several advantages when compared to existing eye-tracking research. Most notably, our method only takes one minute of eye-tracking, a substantial decrease in recording time when compared to about 10 minutes required in previous studies [33,34]. While our method requires a substantially shorter amount of time, it is not a replacement for standard clinical assessments. Extensive experiments are necessary before the true clinical utility and usability of our proposed method can be realised.

Our results support other studies [11–13] that found a significant difference in the overall attention towards geometric stimuli between ASD and TD participants. This significant difference in visual attention was also found between ASD children with severe symptoms and TD children in our study. Despite these differences, using the ratio of visual attention towards the geometric stimuli and the total overall attention and implementing a thresholding technique employed previously [11–13] resulted in lower classification performance than our proposed model. Using an ML-based approach on handcrafted features [23,24] also resulted in lower accuracy in ASD prediction and a similar accuracy in symptom severity prediction than our proposed model. Overall, our results demonstrate the feasibility of using our approach in accurately identifying ASD children and children with severe symptoms. Our model achieved promising performance with high accuracy, sensitivity and specificity.

Finally, most published research reviewed in this paper attempted to identify adults with ASD or older ASD children. In contrast, we investigated the possibility of diagnosing autism and predicting the level of ASD-related symptoms in preschool children (around 4 years old), an age range where diagnosis and assessment are typically performed. As a result, we provide an alternative to augment (and not replace) existing clinical observation tools with a more objective and efficient approach to ASD diagnosis. This takes us closer to an early ASD screening system and allows children to access intervention for better health outcomes. While our results are promising, our proposed approach needs to be trained and tested on a much larger dataset before it can be utilised in clinical settings.

From a clinical perspective, our findings suggest that eye-tracking technology could be used as a biomarker of the presence of ASD and symptom severity in preschool children. Initial findings already found significant correlations between changes in eye-tracking measures and changes in clinical measures captured before and after interventions, suggesting that eye-tracking can be utilised to quantify treatment response [91]. Given the rapid advances in technology supported by the promising performance of the classification models reviewed in this paper, it is not hard to imagine that future research would explore the use of a similar eye-tracking paradigm in predicting other clinical phenotypes and treatment response outcomes in preschool ASD children. This will have a tremendous impact on targeting interventions that maximise health outcomes in patients.

## Limitations

Despite the utility of the current study, there are several limitations to keep in mind. First, there was a gender skew towards males in the ASD group, as would be clinically expected. Nevertheless, further studies with more female participants are required to clarify our results, as differences in autism presentation and diagnosis between males and females have been

documented [92]. For example, studies have shown that girls on the spectrum behave similarly to neurotypical boys and girls on certain socially orientated tasks, such as enhanced attention to faces during scenes that do not have social interactions [93,94]. In addition, TD men with high ASD traits exhibit worse accuracy of gaze shifts, while TD women have similar gaze-following behaviour regardless of ASD traits [95].

Further, the participant groups also differed in sample size, with the ASD group being three times as large as the TD group. The ASD participants in this study were recruited from an ASD-specific centre and there was good uptake to the study. Despite significant efforts of the team to recruit control participants, there was less interest from the families of neurotypical children to participate in the study, which is probably not surprising given the study is less meaningful for children without a developmental diagnosis. We also acknowledge that the dataset size is relatively small in comparison to the dataset required to train modern DL models. To aid our model training and leverage transfer learning, we utilised one of the best dynamic saliency detection model [88] and finetuned its weights to our dataset. This allowed our model to learn better and extract more robust and semantically meaningful features when compared to a model trained from scratch on our dataset. We believe that using the leave-one-out cross-validation approach to train and test the model addressed the class imbalance and small sample size in our study. This validation approach has been used extensively in prior research [14,33,34,43,68,69].

It is also useful to note that the participant groups were matched on chronological age but not on developmental abilities. Further studies with larger sample sizes with a developmentally age-matched group are suggested to confirm our findings. As reported in the Materials and methods section, children with ASD were not excluded from the study if they had a comorbid diagnosis. Although this has implications for any strict interpretation of the findings reported here, the inclusion of comorbid conditions in ASD research is ecologically valid. Indeed, it is rare in clinical practice to encounter a young person who has a 'pure' autism spectrum diagnosis with no other psychiatric or developmental comorbidities.

Finally, we cannot report on the performance of the stimuli-based classification approaches and compare it with our dynamic stimuli-based classification approach since this study is part of a larger study that aimed to find differences in eye-tracking data between ASD and TD participants while watching dynamic stimuli. As such, no eye-tracking data from the same participants were collected while viewing static stimuli.

## Acknowledgments

We extend our gratitude to the children and their families who participated in this study and to the staff where this study was conducted.

## Author Contributions

**Conceptualization:** Ryan Anthony J. de Belen.

**Data curation:** Ryan Anthony J. de Belen, Valsamma Eapen.

**Formal analysis:** Ryan Anthony J. de Belen.

**Funding acquisition:** Valsamma Eapen.

**Investigation:** Ryan Anthony J. de Belen.

**Methodology:** Ryan Anthony J. de Belen.

**Project administration:** Ryan Anthony J. de Belen.

**Software:** Ryan Anthony J. de Belen.

**Supervision:** Tomasz Bednarz, Arcot Sowmya.

**Validation:** Ryan Anthony J. de Belen.

**Visualization:** Ryan Anthony J. de Belen.

**Writing – original draft:** Ryan Anthony J. de Belen.

**Writing – review & editing:** Ryan Anthony J. de Belen, Valsamma Eapen, Tomasz Bednarz, Arcot Sowmya.

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
