## [Decision Letter · Decision Letter 0]

9 May 2023

PONE-D-23-05312Using visual attention estimation on videos for automated prediction of Autism Spectrum Disorder and symptom severity in preschool childrenPLOS ONE

Dear Dr. de Belen,

Thank you for submitting your manuscript to PLOS ONE. After careful consideration, we feel that it has merit but does not fully meet PLOS ONE’s publication criteria as it currently stands. Therefore, we invite you to submit a revised version of the manuscript that addresses the points raised during the review process. You will see that reviewers agree on two main issues:1- Previously published relevant literature needs to be better reported. You need to better position your manuscript with respect to the state-of-the-art. As such, your study goals needs to be more clearly defined.2- The sample size is small and imbalanced, which implies you should pay extra-care about the validation of your model. The leave-one-out technique might not be the best one in this case. Furthermore, I would require that you slightly improve your figures and expand their captions:Figure 1a and 2a: does the x-axis represent fixation duration or number of fixations? Please expend the caption, detail what each panel represents.Figure 3 and 4: please explain in the caption what %Geo values represent and detail the different elements of the box plots (horizontal line = median? Upper and lower whiskers = IQR?). Maybe showing the distribution of the individual datapoints would help, because they are currently not very visible, especially the ones in the box.

We look forward to receiving your revised manuscript.

Kind regards,

Antoine Coutrot

Academic Editor

PLOS ONE

Journal Requirements:

"VE received funding from the Cooperative Research Centre for Living with Autism (Autism CRC) for the data collection"

"NO authors have competing interests"

5. We note that Figures 5, 6, 7 and 8 in your submission contain copyrighted images. All PLOS content is published under the Creative Commons Attribution License (CC BY 4.0), which means that the manuscript, images, and Supporting Information files will be freely available online, and any third party is permitted to access, download, copy, distribute, and use these materials in any way, even commercially, with proper attribution. For more information, see our copyright guidelines: http://journals.plos.org/plosone/s/licenses-and-copyright.

a. You may seek permission from the original copyright holder of Figures  5, 6, 7 and 8 to publish the content specifically under the CC BY 4.0 license. 

Reviewers' comments:

Reviewer's Responses to Questions

**Comments to the Author**

1. Is the manuscript technically sound, and do the data support the conclusions?

Reviewer #1: Partly

Reviewer #2: Partly

Reviewer #3: Partly

2. Has the statistical analysis been performed appropriately and rigorously? 

Reviewer #1: Yes

Reviewer #2: I Don't Know

Reviewer #3: Yes

3. Have the authors made all data underlying the findings in their manuscript fully available?

Reviewer #1: No

Reviewer #2: Yes

Reviewer #3: No

4. Is the manuscript presented in an intelligible fashion and written in standard English?

Reviewer #1: Yes

Reviewer #2: Yes

Reviewer #3: Yes

5. Review Comments to the Author

Reviewer #1: The following are the major issues I found with the paper:

The paper is very hard to follow, it is very jumbled.

1. The paper is not laid out in a standard sections: Limitations and Results are before Methods. It is generally Methods, Results, Limitations. I did not understand why the sections were jumbled, made it difficult to understand the paper.

2. References are very haphazard. Some citations have all the authors listed, some end in et al. Kindly stick to one of the two.

3. No base line results. The paper provides no baseline results on their dataset from previous studies. A table that shows previous methods results and the paper's approach will be nice and will help catch the attention of the reader.

Previous methods reported a performance of 93.45% and the proposed approach is 94.47% without a comparison on same dataset following the same cross validation protocols, it is difficult to say if it was the method, the data or just a statistical anomaly.

Reviewer #2: Thank you for the opportunity to review the manuscript “Using visual attention estimation on videos for automated prediction of Autism Spectrum Disorder and symptom severity in preschool children”.

The study aimed to predict and stratify ASD using visual attention estimation. I commend the authors for their interest in autism and innovative computational models. I think this study could be interesting for the scientific community, particularly for the stratification model based on visual attention. However, I cannot recommend this manuscript for publication in its current form and I suggest major revisions. I hope that the following comments will assist the authors in their revision process.

- In the introduction section, authors claim that “there is a lack of diagnostic paradigms that leverage dynamic stimuli” (line 68); they “go beyond static visual stimuli and utilize dynamic visual stimuli for eye tracking experiments” (lines 70-71), and their approach “provides an extension from static stimuli, widening the diagnostic paradigm to include dynamic stimuli” (lines 165-166). It seems that the use of dynamic visual stimuli in the computational identification of ASD is something novel that it is introduced here. Nevertheless, there are previous instances of machine learning applications on visual attention towards dynamic stimuli (see the contribution of Minissi et al. (2022) in which four studies using dynamic stimuli are presented. https://link.springer.com/article/10.1007/s10803-021-05106-5), and authors used the same dynamic stimulus of Moore at al. (2018) in which a computational identification model of ASD is reported.

In addition, in contrast to the idea that using dynamic stimuli is somehow “novel”, in line 122 authors described some literature using dynamic stimuli and visual attention for ASD classification; thus, which is the novelty of the present study regarding the type of stimulus? I suggest authors define clearly how this study differs from previous literature using the same type of social stimuli and which is the novelty.

- From lines 103 to 150, a brief state of the art regarding ASD classification based on static and dynamic stimuli is presented. It is essential to give an idea to the reader about previous literature. I suggest authors discuss these studies more deeply rather than just mention them. The relation between the present study and the previous ones needs to be clarified.

- Be careful with terms in the paragraph at line 151: the risk of ASD is not the same as ASD symptom severity. The comparison scores of the ADOS (used in the present study) give an idea of the severity of ASD instead of the risk of ASD, and in this paragraph, some studies assessing the risk of ASD are presented.

- Line 194: it is surprising finding a comparison between the present model and those of Pierce et al. (2011) and Moore et al. (2018) since the authors should have reported in the introduction that this would have been a study purpose. I suggest the authors revise the introduction section to define study goals clearly.

- The discussion is short and leaves the reader to intend that using dynamic stimuli is a novelty of this study, but it is not. The discussion should present study results, their comparison with previous literature, and the step forward that the study made. How the current model improves ASD identification compared to previous models? How do the findings relate to Minissi et al. (2022) and Kollias et al. (2021) reviews?

- We could say that unequal groups and reduced sample sizes are the major limitations of this study, and findings should be interpreted cautiously. In line 269 authors claimed that the LOOCV approach is suitable for their model and addresses the group imbalance. However, for my little knowledge of data science, variant of k-fold cross-validation are recommended for evaluating the performance of a model when groups are imbalanced, and the sample size are reduced. Stratified k-fold cross-validation ensures that each fold of the cross-validation process has roughly the same proportion of samples from each group as the original dataset. This approach is particularly useful when dealing with imbalanced datasets or unequal group sizes. In contrast, LOOCV may not be suitable for small datasets with unequal group sizes because it requires leaving out a single data point at each iteration, which can result in too few data points in the training set, making it difficult to obtain a reliable estimate of the model performance. Therefore, in this scenario, stratified k-fold cross-validation can be a good option as it strikes a balance between the computational efficiency of k-fold cross-validation and the need to account for group imbalances in the dataset. As reported in line 270, a similar study used 5-fold cross-validation to address the class imbalance. Still, the authors should have discussed why they used LOOCV compared to other strategies.

- Lines 318-322: Please deepen the description of how the experimenter calibrated the eye tracking system with young children (4 yo) with high-moderate severity of ASD.

Minor comments

- In the result section, if I got it right, figure 1 and 2 the model metrics are presented as the number of fixations increased, but in the horizontal axis of both plots “fixation length” is reported.

- How symptom severity has been measured should be reported in the participant section. Some information is unclear and presented in lines 411-413.

- In the limitation section, it is reported that there was a gender imbalance in the sample; however, this is not reported in the participant section. In addition, the limitation section is long, and some of the info may be moved to further sections of the paper.

Reviewer #3: The authors present a machine-learning-based model for automatic screening of ASD and for severity (two levels) prediction, which considers features extracted from dynamic visual stimuli. The reported experimental results show promising performance and comparable to the performance reported by related works on static stimuli.

This paper addresses a very relevant and timely topic, which, although a is focus of a very active research work lately, still requires more studies, models and datasets. Therefore, I think that the paper can provide useful insights for the research community in the field. However, I'm concerned about the limited validation of the proposed approach, which I'm not sur if it is sufficient for a journal publication. In this sense, my major concern is that the proposed approach has only been validated using a leave-one-out technique. Although it is a common and valid practice, I would expect a deeper validation of models addressed to applications like this, based on the use of separate training and testing datasets. Several works have reported a significant decrease of the performance of the models when using a "secret" testing dataset in comparison with leave-one-out methods. I'm aware of the difficulty of doing this, given the lack of publicly available datasets, but probably the authors can report some more results splitting their dataset in two subsets. In addition, the proposed approach has been compared with a simple thresholding approach [10-12], but they have not compared their model with any pf the existing ones for dynamic stimuli. I'm also aware of the lack of open-source models, but in case the authors have identified any, it would be nice to compare the performance of their approach with those.

In this sense, if the authors have identified any publicly available dataset (e.g., Carette and colleagues have recently published one) it would be nice if they can use it to further test their model. If not, at least, I think that a review of existing datasets (probably as part of the related work section) would be beneficial to the readers.

Some other comments are:

- In line 162-163, the authors say "limited research has been performed to explore the effectiveness of temporal information captured during dynamic stimuli viewing". It would be nice to elaborate on this and provide more details about what do the authors mean with it. I would expect from almost all the related works mentioned by the authors working with dynamic stimuli to exploit the temporal information that this stimuli provide (in contrast to using static ones).

- I may have missed it, but I have not seen the specific numbers related to the gender of the participants in the test (although the authors mentioned that the pool of participants was unbalanced). Also in this sense, although it was mentioned as a limitation by the authors, the unbalance of the dataset in terms of children with ASD and with TD is an important flaw.

- In terms of the stimuli used, although the detail information can be found in the papers [10-11], for self-contention of this paper, it would be nice that the authors provide more info of them. For example, which was the duration of the scenes? How were they presented to the participants (e.g., in random order)? Were the videos played in full-screen? Which was the resolution of the videos?...

- In line 335, the authors say that they have used "An additional data quality assessment... to determine the overall nature of the visual attention of the participants to the stimuli", but no specific information has been provided about it.

- In line 339, the authors state that "An independent-samples t-test was used to investigate differences in visual attention across two groups for diagnosis (ASD vs. TD) and severity prediction (moderate vs. severe)". However, no results are reported or no indications on were to finde them are included.

- In the expression in line 375, how the coefficients (0.1) that apply to L_CC and L_NSS were obtained?

Finally, the paper is, in general, well written, but a careful proof-read would be beneficial to avoid some typos and minor errors. For example, in the introduction, line 59 "Pierce et al." appears twice; line 104 "to an autism diagnosis" may be "to autism diagnosis"; please check for extra or missing spaces before or after commas; in lines 276 and 277 authors use "comorbid" and "co-morbid"; etc.

6. PLOS authors have the option to publish the peer review history of their article (what does this mean?). If published, this will include your full peer review and any attached files.

Reviewer #1: No

Reviewer #2: No

Reviewer #3: No

---

## [Author Response · Author response to Decision Letter 0]

24 Jul 2023

We thank the reviewers for taking their valuable time in reviewing our manuscript. We are very pleased to see that all reviewers found our manuscript to be valuable and/or suggested further improvements to our work. We agree with all the points raised and made revisions accordingly. We believe that they have greatly improved our submission.

We also thank you for providing us with an opportunity to submit a revision and clarify any concerns raised by the reviewers. To address these concerns, we made the following:

Major changes:

1. We have expanded the Related Works section with the inclusion of Tables 2,3,4 and 5 for a more detailed comparison of published works. More specifically, each table contains the mean age of the participants, gender distribution, stimuli and input used, methodology and conclusion. This allowed us to better situate our work in the literature, as well as highlight our work’s novelty.

2. We have added other performance comparisons with a prior approach that computed eye tracking variables and compared different machine learning models for comparison)1,2.

3. We have included a new section that provides a summary of different publicly available datasets for classification. This contains Table 1 that lists information about the reference, their target application area, the mean age of the participants, sample size, stimuli used and data format provided by the authors.

Minor changes:

1. We have revised the manuscript format to meet PLOS ONE’s style requirements.

2. We have provided more information about the gender distribution, eye-tracking procedures and clinical measures.

3. We have made improvements to the figures by providing more information in their corresponding captions.

4. Removal of the snapshot of the copyrighted stimuli.

---

## [Decision Letter · Decision Letter 1]

18 Dec 2023

Using visual attention estimation on videos for automated prediction of Autism Spectrum Disorder and symptom severity in preschool children

PONE-D-23-05312R1

Dear Dr. de Belen,

We’re pleased to inform you that your manuscript has been judged scientifically suitable for publication and will be formally accepted for publication once it meets all outstanding technical requirements.

Kind regards,

Antoine Coutrot

Academic Editor

PLOS ONE

Additional Editor Comments (optional):

Reviewers' comments:

Reviewer's Responses to Questions

**Comments to the Author**

1. If the authors have adequately addressed your comments raised in a previous round of review and you feel that this manuscript is now acceptable for publication, you may indicate that here to bypass the “Comments to the Author” section, enter your conflict of interest statement in the “Confidential to Editor” section, and submit your "Accept" recommendation.

Reviewer #2: (No Response)

2. Is the manuscript technically sound, and do the data support the conclusions?

Reviewer #2: Yes

3. Has the statistical analysis been performed appropriately and rigorously? 

Reviewer #2: Yes

4. Have the authors made all data underlying the findings in their manuscript fully available?

Reviewer #2: No

5. Is the manuscript presented in an intelligible fashion and written in standard English?

Reviewer #2: Yes

6. Review Comments to the Author

Reviewer #2: The manuscript has improved in terms of structure. Now, the objectives of the study are clearly presented, making it easier to follow. Here are some minor comments that I believe could further enhance the manuscript:

• The introduction is excessively long

• I don't understand why ASD risk is reported in the related works since it is not an objective of the study.

• Participants' ages should be reported in the appropriate section rather than in the results.

• To the best of my knowledge, if the used stimulus is composed of images, it is considered a static stimulus instead of a dynamic one. Please discuss why the used stimulus can be considered dynamic instead of static.

• Some ref in the text present formatting errors

7. PLOS authors have the option to publish the peer review history of their article (what does this mean?). If published, this will include your full peer review and any attached files.

Reviewer #2: No

---

## [Editor Report · Acceptance letter]

3 Feb 2024

PONE-D-23-05312R1 

PLOS ONE

Dear Dr. de Belen, 

I'm pleased to inform you that your manuscript has been deemed suitable for publication in PLOS ONE. Congratulations! Your manuscript is now being handed over to our production team.

Kind regards, 

on behalf of

Dr. Antoine Coutrot 

Academic Editor

PLOS ONE